



# Modeling reactive ammonia uptake by secondary organic aerosol in CMAQ: application to continental US

Shupeng Zhu[1], Jeremy R. Horne[1], Julia Montoya-Aguilera[2], Mallory L. Hinks[2], Sergey A. Nizkorodov[2], and Donald Dabdub[1]

[1]Computational Environmental Sciences Laboratory, Department of Mechanical & Aerospace Engineering, University of California, Irvine, Irvine, CA, 92697-3975, USA
[2]Department of Chemistry, University of California, Irvine, Irvine, CA, 92697-3975, USA

*Correspondence to:* Donald Dabdub (ddabdub@uci.edu)

**Abstract.** Ammonium salts such as ammonium nitrate and ammonium sulfate constitute an important fraction of the total fine particulate matter ($PM_{2.5}$) mass. While the conversion of inorganic gases into particulate phase sulfate, nitrate, and ammonium is now well understood, there is considerable uncertainty over interactions between gas-phase ammonia and secondary organic aerosols (SOA). Observations have confirmed that ammonia can react with carbonyl compounds in SOA, forming nitrogen-

containing organic compounds (NOC). This chemistry can reduce gas-phase $NH_3$ concentration and therefore affect the amount of ammonium nitrate and ammonium sulfate in particulate matter (PM). In order to investigate the importance of such reactions, a first-order loss rate for ammonia onto SOA was implemented into the Community Multiscale Air Quality (CMAQ) model based on the ammonia uptake coefficients reported in the literature. Simulations over the continental US were performed for the winter and summer of 2011 with a range of uptake coefficients ($10^{-3}$ - $10^{-5}$). Simulation results indicate that a

significant reduction in gas-phase ammonia is possible due to its uptake onto SOA; domain-averaged ammonia concentrations decrease by 31.3 % in the winter, and 67.0 % in the summer with the highest uptake coefficient ($10^{-3}$). As a result, the concentration of particulate matter is also significantly affected, with a distinct spatial pattern over different seasons. PM concentrations decreased during the winter, largely due to the reduction in ammonium nitrate concentrations. On the other hand, PM concentrations increased during the summer due to increased production of biogenic SOA production resulting from

enhanced acid-catalyzed uptake of isoprene-derived epoxides. While ammonia emissions expected to increase in the future, it is important to include $NH_3$ + SOA chemistry in air quality models.

## 1 Introduction

As the most abundant basic gas in the atmosphere (Behera et al., 2013), gaseous ammonia ($NH_3$) has long been considered responsible for controlling the eutrophication and acidification of ecosystems (Sutton et al., 1993; Erisman et al., 2008; Sheppard

et al., 2011). More recently, studies also demonstrated the importance of ammonia in the formation of airborne fine particulate matter ($PM_{2.5}$) (West et al., 1999; Vayenas et al., 2005; Wang et al., 2013). Through reactions with acidic species, ammonia is converted into ammonium salts, such as ammonium nitrate and ammonium sulfate, which constitute an important fraction of total $PM_{2.5}$ mass (Behera and Sharma, 2010). These aerosols have been proven to impact human health (Pope III et al.,



2002; Lelieveld et al., 2015), visibility (Ye et al., 2011) and the atmospheric radiative balance (Xu and Penner, 2012; Park et al., 2014). In the US, the largest ammonia emission source is agricultural activity ( 85% of total US ammonia emissions) (Pinder et al., 2004, 2006), largely from animal waste and commercial fertilizer application, such as the intensive farming in California's central valley (Jovan and McCune, 2005) and industrialized hog farms in central North Carolina (McCulloch

et al., 1998; Aneja et al., 2000). The ammonia rich plumes from those areas drive most of the nitric acid into the particle phase, resulting in high $PM_{2.5}$ concentrations in those regions (Neuman et al., 2003; Baek and Aneja, 2004). Recent studies have also shown that atmospheric ammonia has increased during the last two decades, a trend that is expected to continue as a result of global warming, increasing agricultural activity and intensifying fertilizer use due to growing population (Galloway et al., 2008; Amann et al., 2013; Warner et al., 2017).

While the conversion of inorganic gases into particulate phase sulfate, nitrate, and ammonium is now fairly well understood (Seinfeld and Pandis, 2016), there is considerable uncertainty over interactions between gas phase ammonia and organic compounds in secondary organic aerosols (SOA). Laboratory studies have shown that ammonia can react with SOA compounds in two ways. It can either react with organic acids to form ammonium salts (Na et al., 2007), or participate in reactions with certain carbonyl compounds forming heterocyclic nitrogen-containing organic compounds (Updyke et al., 2012; Laskin et al., 2015).

In addition, a browning effect on SOA under $NH_3$ exposure is observed by Updyke et al. (2012), indicating the production of light-absorbing products. These processes are not included in current air quality models, which could lead to over estimation of gaseous ammonia concentrations, and thus inorganic aerosol concentration. Additionally, the neglect of these two processes may also result in under estimation of organics aerosol, especially species related to acid catalyzed reactions (Lin et al., 2013) and in incorrect prediction of aerosol particle acidity.

Recently, chemical uptake coefficients for ammonia onto SOA were reported for the first time by Liu et al. (2015). Those coefficients were on the order of $\sim 10^{-3}$-$10^{-2}$ for fresh SOA, decreasing significantly to $< 10^{-5}$ after 6h of reaction. They observed that the nitrogen-containing organic compounds (NOC) mass contributed $8.9 \pm 1.7$ and $31.5 \pm 4.4$ wt% to the total $\alpha$-pinene and $m$-xylene-derived SOA, respectively, and 4-15 wt% of the total nitrogen in the system.

In this work, we investigate the impact of ammonia uptake by SOA on $PM_{2.5}$ and $NH_3$ concentrations, by implementing a
first-order loss rate for ammonia onto SOA into the Community Multiscale Air Quality (CMAQ) modeling system based on ammonia uptake coefficients reported by Liu et al. (2015). Air quality simulations over the continental US were performed with a range of uptake coefficients to determine the sensitivity of $PM_{2.5}$ and $NH_3$ concentration to the magnitude of the uptake coefficient. Furthermore, in order to investigate the seasonal impact on this process, simulations were conducted for both winter and summer. The modeling method used in this analysis will first be presented in section 2. Then, simulation results will be
analyzed based on both observational data and sensitivity comparisons between different scenarios in section 3. Finally, in section 4, the importance of including this process in air quality models will be discussed.



## 2  Methodology

The CMAQ modeling system (Byun and Schere, 2006) is a widely used state-of-the-art chemical transport model. In the United States, it is among the most commonly used air quality models in attainment demonstrations for National Ambient Air Quality Standards for ozone and $PM_{2.5}$ (USEPA, 2007). In this study, eight simulations were conducted using the latest 2017 release

of CMAQ (Version 5.2), including one base case simulation for the winter (Jan. 1 - Feb. 27, 2011), one base case simulation for the summer (Jul. 1 - Aug. 30, 2011), and three different $NH_3$ uptake scenarios for each period. The Carbon Bond version 6 (CB6) mechanism (Yarwood et al., 2010) was used for the gas-phase chemistry, which includes 127 species as detailed on the website (Adams, 2017), and the AERO6 module was used for aerosol dynamics, which includes 21 inorganic species and 34 organic species (28 SOA and 6 primary organic species) as detailed on the CMASWIKI website (Pye, 2016). The modeling

domain used in this study covers the contiguous US using a 12 km × 12 km horizontal grid resolution (resulting in 396 (x) × 246 (y) = 97,416 grid cells) and a 29-layer logarithmic vertical structure (set on a terrain following sigma coordinate, from the surface to 50 hPa) with the depth of the first layer around 26 m. Only the simulation results from the first layer, representative of ground level, were used for the analysis in this study.

The meteorological fields were derived from NCEP FNL (Final) Operational Global Analysis data (NCEP, 2000) using the

Weather Research and Forecasting Model (WRF, version 3.7) (Skamarock et al., 2008), with the MODIS land use database (Friedl et al., 2010) and the YSU parametrization (Hong et al., 2006) for the planetary boundary layer. The WSM3 scheme (Hong et al., 2004) was used for the microphysics option of WRF, and the Kain - Fritsch convective parametrization (Kain, 2004) was used for cumulus physics. These fields were then processed using Version 4.3 of Meteorology Chemistry Interface Program (MCIP) (Otte and Pleim, 2010). The initial and boundary conditions were obtained from the Model for OZone

And Related chemical Tracers (Mozart v2.0) (Horowitz et al., 2003). Emissions were generated based on the 2014 National Emissions Inventory (NEI) (EPA, 2017a) and processed by the Sparse Matrix Operator Kernel Emission (SMOKE, version 4.5) processor (EPA, 2017b). Biogenic emissions were obtained from the Biogenic Emission Inventory System (BEIS) (Pierce and Waldruff, 1991), and emissions from cars, trucks, and motorcycles were calculated with MOBILE6 (EPA, 2003).

In this study, the AERO6 module in CMAQ was updated to simulate the heterogeneous uptake of $NH_3$ by SOA. AERO6 used

the modal representation to simulate aerosol dynamics (Binkowski and Roselle, 2003). The size distribution of the aerosols are represented by 3 log-normal modes: the Aitken mode (size up to approximately 0.1 $\mu$m), the accumulation mode (size between 0.1 $\mu$m to 2.5 $\mu$m) and the coarse mode (size between 2.5 $\mu$m to 10 $\mu$m). In the AERO6 modal approach, three integral properties of the size distribution are followed for mode $j$: the total particle number concentration $N_j$, the total surface area concentration $S_j$, and the total mass concentration $m_{ij}$ of each individual chemical component $i$. In order to calculate the

total uptake of $NH_3$ by SOA, one must know the representing surface area concentration of SOA ($S_{SOA}$), that can be calculated as follows (assuming unified density amount different chemical components):

$$S_{SOA} = \sum_{j=1}^{x} (S_j \times \frac{\sum_{i=1}^{y} m_{ij}}{\sum_{k=1}^{z} m_{kj}}) \tag{1}$$



where $y$ is the total number of SOA species in mode $j$, $z$ is the total number of aerosol species in mode $j$, and $x$ is the total number of modes that contain SOA species. Here, $x=2$ since SOA only exist in the Atiken mode and the accumulation mode. From $S_{SOA}$ the first order rate of $NH_3$ uptake can be calculated as:

$$k = \gamma \times \frac{v_{NH_3} \times S_{SOA}}{4} \tag{2}$$

where $\gamma$ is the reactive uptake coefficient for ammonia, and $v_{NH_3}$ is the average speed of $NH_3$ molecules (609 $m/s$ at 298 K). The above calculations were performed separately for each grid cell at every time step to obtain the effective first-order rate constant for each individual cell at each time step. The first-order rate constant of $NH_3$ uptake was then multiplied by the gas-phase $NH_3$ concentration to determine the loss rate of $NH_3$ in each cell at each time step.

The process responsible for the chemical uptake of ammonia into particles is not expected to significantly change the mass
concertation of particulate organics. In this reaction, the carbonyl group of an SOA compounds is converted into an imine group and a molecule of water is produced as a by-product. The imine product can further react by an intermolecular cyclization to produce heterocyclic organic compounds, with a loss of an additional water molecule (Laskin et al., 2014). The difference in molecular weights of two $H_2O$ molecules and one $NH_3$ molecule ($2 \times 18$ - 17 = 19 $g/mol$) is small relative to a molecular weight of a typical SOA compounds (about 200 $g/mol$). Therefore, for the sake of simplicity, we neglected the loss of the
mass of particulate organics mass in this simulation. This assumption is supported by experimental observations described by Horne et al. (2017), in which SOA particles exposed to ammonia in a smog chamber did not change their size distribution but showed clear evidence of incorporation of organic nitrogen into the particles in on-line and off-line mass spectra.

The ammonia uptake coefficients ($\gamma$) used in this study were determined by considering the values reported in the work of Liu et al. (2015), as well as the maximum possible uptake based on the available SOA particles. Liu et al. (2015) reported
a range of possible uptake coefficient from $10^{-5}$ to $10^{-2}$. However, some of our initial modeling tests showed that the use of $10^{-2}$ uptake coefficient value would lead to an unrealistic amount of $NH_3$ taken up by SOA, where within a single time step, the number of moles of $NH_3$ taken up exceeded 10% of the total moles of SOA in one grid cell. Experiments (Liu et al., 2015; Horne et al., 2017) suggest that only about 10% of SOA molecules can react with $NH_3$ to form nitrogen-containing organic compounds (NOC). Additionally, in the study of Liu et al. (2015), the uptake coefficients are measured based on only
a few SOA species (SOA formed from ozonolysis of $\alpha$-pinene and OH oxidation of $m$-xylene); other SOA species might not have the same properties. Furthermore, the highest value of uptake coefficient was only observed at the initial period of the experiment of Liu et al. (2015) and decreased rapidly over time. Based on the considerations above, uptake coefficient of $10^{-3}$ was considered a more reasonable upper limit value for our application instead of $10^{-2}$. Thus, four simulations were performed for each period to investigate the sensitivity of $NH_3$ removal to changes in the uptake coefficient: (a) base case with no $NH_3$
uptake, (b) $NH_3$ uptake with $\gamma = 10^{-3}$, (c) $NH_3$ uptake with $\gamma = 10^{-4}$, (d) $NH_3$ uptake with $\gamma = 10^{-5}$.

Results from each simulation were evaluated by comparing with observations from multiple monitoring networks. Then simulation results for scenario (b), (c) and (d) are compared to the base case results in (a) to determine the impact of different uptake coefficients on different gas and particle phase species. The value of $\gamma$ was assumed to remain constant in each scenario (i.e., no saturation or aging effects), which means each scenario represents an upper limit for the amount of $NH_3$ that would



be taken up by SOA with the chosen value of the uptake coefficient. No further changes were made to the model or its inputs between each scenario. Results of the first 7 days of each simulations were discarded as a model spin up period to minimize the effect of initial conditions and allow sufficient time for $NH_3$ removal process to occur.

## 3 Results and Discussion

### 3.1 Model validation

First, base case simulation results of $PM_{2.5}$, $PM_{10}$ and $O_3$ are compared with the observations from the U.S. Environmental Protection Agency's Air Quality System (AQS) to evaluate the model performance. The AQS network (https://www.epa.gov/aqs) is geographically diverse and spans the entire US. It is also an excellent source of quality assured measurements, with hourly recorded concentrations for $PM_{2.5}$, $PM_{10}$ and $O_3$. The definitions of the statistical parameters used in this study are detailed in the supporting information (SI) (Table S1).

Table 1 shows good model performance for $O_3$, as the statistics meet the recommended performance criteria (|MNGB| $\leq 15\%$ and MNGE $\leq 30\%$) (Russell and Dennis, 2000). Only the two base cases simulations are shown in Table 1, because the change in $NH_3$ uptake coefficient has no impact on $O_3$. Table 2 shows the statistics for $PM_{2.5}$ for both the summer and winter. Cases satisfied the model performance criteria proposed by (Boylan and Russell, 2006) with MFE $\leq 75\%$ and |MFB| $\leq 60\%$. The model performance for winter is much better than for the summer, as the amount of $PM_{2.5}$ is overestimated during the summer. The impact of different $NH_3$ uptake coefficients on $PM_{2.5}$ is also reflected in the statistics. For the winter, increasing the $NH_3$ uptake coefficient leads to a decrease of the total $PM_{2.5}$ and a slightly better model performance when compared to the observations. On the contrary, larger $NH_3$ uptake coefficients cause higher $PM_{2.5}$ concentration during the summer, resulting in a larger discrepancies compared with measurements. The reasons for such seasonal differences will be analyzed in section 3.2.4. The statistics of $PM_{10}$ show much closer agreement between the simulation results and the observations than $PM_{2.5}$, as shown on Table S2 in the SI. The MFE is similar to that of $PM_{2.5}$, while much smaller MFB values are found for the summer. Similar to $PM_{2.5}$, the increase of $NH_3$ uptake coefficient leads to lower $PM_{10}$ concentration for the winter, but higher $PM_{10}$ concentration for the summer. One possible explanation for the different performance between $PM_{2.5}$ and $PM_{10}$ could be the underestimation of coarse mode particle due to the mode-species limitation of CMAQ. Most of the SOA species are not allowed to grow into the coarse mode and their mass could be trapped in the accumulation mode therefore cause this overestimation.

Second, the simulated concentration of gas-phase $NH_3$ is compared to observation data from the Ammonia Monitoring Network (AMoN). In each AMoN site, samples are deployed for 2-week periods. Details about the network and its sampling method can be found on NADP (2014). Table 3 shows the statistics between each simulation case and the measurement data. The seasonal influence is quite clear in the statistics of the two base case simulations. Similar to the $PM_{2.5}$, the model overestimates the $NH_3$ concentration for the summer. On the contrary, the simulated $NH_3$ concentration is underestimated for the winter. The impacts of different $NH_3$ uptake coefficients on $NH_3$ concentrations are consistent between the winter and the





**Table 1.** Comparison between the base case simulation results for $O_3$ and observations from the AQS network. (Obs. stands for observation. Sim. stands for simulation. Corr. stands for correlation, No. Sites means number of observation site used for statistics.)

|  | Obs. mean | Sim. mean | RMSE | Corr. | MNGB | MNGE | No. Sites |
|---|---|---|---|---|---|---|---|
| Period | ppb | ppb | ppb | % | % | % | |
| Summer | 41.1 | 50.9 | 16.7 | 56.7 | 12.0 | 29.7 | 1262 |
| Winter | 27.3 | 33.9 | 10.4 | 51.4 | 8.8 | 23.1 | 664 |

**Table 2.** Comparison between simulation results for $PM_{2.5}$ and observations from the AQS network. (Obs. stands for observation; Sim. stands for simulation. Corr. stands for correlation; No. Sites means number of observation site used for statistics.)

| Scenario | Period | Obs. mean $\mu g/m^{-3}$ | Sim. mean $\mu g/m^{-3}$ | RMSE $\mu g/m^{-3}$ | Corr. % | MFB % | MFE % | No. Sites |
|---|---|---|---|---|---|---|---|---|
| Base | Summer | 12.6 | 21.9 | 18.1 | 17.8 | 36.7 | 62.7 | 176 |
| $\gamma=10^{-3}$ | Summer | 12.6 | 24.1 | 20.5 | 18.3 | 41.2 | 66.3 | 176 |
| $\gamma=10^{-4}$ | Summer | 12.6 | 22.1 | 18.4 | 17.8 | 37.2 | 63.1 | 176 |
| $\gamma=10^{-5}$ | Summer | 12.6 | 21.9 | 18.1 | 17.8 | 37.0 | 62.9 | 176 |
| Base | Winter | 12.3 | 13.0 | 11.4 | 31.3 | 2.8 | 60.9 | 166 |
| $\gamma=10^{-3}$ | Winter | 12.3 | 12.6 | 11.1 | 31.4 | 0.6 | 60.4 | 166 |
| $\gamma=10^{-4}$ | Winter | 12.3 | 12.9 | 11.4 | 31.4 | 2.4 | 60.8 | 166 |
| $\gamma=10^{-5}$ | Winter | 12.3 | 13.0 | 11.4 | 31.3 | 2.7 | 60.9 | 166 |

**Table 3.** Comparison between simulation results for $NH_3$ and observations from the AMoN network. (Obs. stands for observation; Sim. stands for simulation. Corr. stands for correlation; No. Sites means number of observation site used for statistics.)

| Scenario | Period | Obs. mean $\mu g/m^{-3}$ | Sim. mean $\mu g/m^{-3}$ | RMSE $\mu g/m^{-3}$ | Corr. % | MFB % | MFE % | No. Sites |
|---|---|---|---|---|---|---|---|---|
| Base | Summer | 1.36 | 2.17 | 1.41 | 20.2 | 46.7 | 72.2 | 46 |
| $\gamma=10^{-3}$ | Summer | 1.36 | 0.63 | 1.07 | -26.1 | -70.1 | 96.4 | 46 |
| $\gamma=10^{-4}$ | Summer | 1.36 | 1.48 | 1.08 | -2.0 | 7.3 | 63.2 | 46 |
| $\gamma=10^{-5}$ | Summer | 1.36 | 1.30 | 1.30 | 18.1 | 38.0 | 68.9 | 46 |
| Base | Winter | 0.77 | 0.37 | 0.57 | 26.2 | -63.3 | 88.7 | 19 |
| $\gamma=10^{-3}$ | Winter | 0.77 | 0.31 | 0.60 | 29.7 | -78.9 | 98.0 | 19 |
| $\gamma=10^{-4}$ | Winter | 0.77 | 0.36 | 0.58 | 27.5 | -65.9 | 90.1 | 19 |
| $\gamma=10^{-5}$ | Winter | 0.77 | 0.37 | 0.57 | 26.5 | -63.6 | 88.9 | 19 |



**Table 4.** Comparison between simulation results for $NH_4^+$ and observations from CSN network. (Obs. stands for observation; Sim. stands for simulation. Corr. stands for correlation; No. Sites means number of observation site used for statistics.)

| Scenario | Period | Obs. mean $\mu g/m^{-3}$ | Sim. mean $\mu g/m^{-3}$ | RMSE $\mu g/m^{-3}$ | Corr. % | MFB % | MFE % | No. Sites |
|---|---|---|---|---|---|---|---|---|
| Base | Summer | 0.82 | 0.98 | 0.70 | 31.8 | 7.7 | 71.3 | 187 |
| $\gamma=10^{-3}$ | Summer | 0.82 | 0.83 | 0.62 | 31.4 | -5.3 | 70.3 | 187 |
| $\gamma=10^{-4}$ | Summer | 0.82 | 0.92 | 0.66 | 32.0 | 3.2 | 70.5 | 187 |
| $\gamma=10^{-5}$ | Summer | 0.82 | 0.96 | 0.69 | 31.9 | 6.8 | 71.1 | 187 |
| Base | Winter | 1.30 | 1.20 | 0.96 | 45.8 | -12.8 | 64.5 | 187 |
| $\gamma=10^{-3}$ | Winter | 1.30 | 1.08 | 0.93 | 45.1 | -21.1 | 64.3 | 187 |
| $\gamma=10^{-4}$ | Winter | 1.30 | 1.18 | 0.95 | 45.6 | -14.1 | 64.4 | 187 |
| $\gamma=10^{-5}$ | Winter | 1.30 | 1.20 | 0.96 | 45.8 | -12.9 | 64.4 | 187 |

**Table 5.** Comparison between simulation results for $NO_3^-$ and observations from CSN network. (Obs. stands for observation; Sim. stands for simulation. Corr. stands for correlation; No. Sites means number of observation site used for statistics.)

| Scenario | Period | Obs. mean $\mu g/m^{-3}$ | Sim. mean $\mu g/m^{-3}$ | RMSE $\mu g/m^{-3}$ | Corr. % | MFB % | MFE % | No. Sites |
|---|---|---|---|---|---|---|---|---|
| Base | Summer | 0.47 | 0.88 | 0.85 | 17.8 | 31.1 | 87.3 | 187 |
| $\gamma=10^{-3}$ | Summer | 0.47 | 0.46 | 0.54 | 14.7 | -38.2 | 90.1 | 187 |
| $\gamma=10^{-4}$ | Summer | 0.47 | 0.70 | 0.68 | 18.2 | 10.3 | 80.6 | 187 |
| $\gamma=10^{-5}$ | Summer | 0.47 | 0.84 | 0.81 | 18.1 | 27.6 | 85.8 | 187 |
| Base | Winter | 2.43 | 3.14 | 2.57 | 40.4 | 31.0 | 75.2 | 187 |
| $\gamma=10^{-3}$ | Winter | 2.43 | 2.74 | 2.29 | 40.0 | 20.5 | 71.0 | 187 |
| $\gamma=10^{-4}$ | Winter | 2.43 | 3.07 | 2.52 | 40.4 | 29.3 | 74.4 | 187 |
| $\gamma=10^{-5}$ | Winter | 2.43 | 3.13 | 2.56 | 40.4 | 30.8 | 75.1 | 187 |

summer, the $NH_3$ concentration decreases as the uptake coefficient increases. However, such impact is much more significant during the summer than the winter.

Finally, simulation results of individual inorganic aerosol compounds (e.g., $NH_4^+$, $SO_4^{2-}$, and $NO_3^-$) are also compared with measurement data obtained from the EPA's Chemical Speciation Network (CSN). The CSN network collect 24-h integrated samples every day (midnight to midnight) of major fine particle chemical components and most of CSN sites are in urban areas. Detailed description of the network and its sampling protocol is described Malm et al. (2004). The statistics for $SO_4^{2-}$ presented on Table S3 in the SI shows good model performance, there is good agreement between mean observed and simulated concentrations with small MFB and MFE values that satisfied the model performance goal proposed by Boylan and Russell (2006) (|MFB| $\leq$ 30% and MFE $\leq$ 50%). The statistics of other scenarios are not presented in the table, as the change of $NH_3$ uptake coefficient shows no observable impact on the $SO_4^{2-}$ statistics. This is due to the extremely low volatility of sulfuric acid,



which forces almost the entire $SO_4^{2-}$ to be condensed into the aerosol phase, regardless the concentration of $NH_3$. For $NH_4^+$ (Table 4), in general, the statistics show a good model performance, as the MFB and MFE satisfied the model performance criteria proposed by Boylan and Russell (2006) in all 8 scenarios. For the summer, the $NH_4^+$ is slightly overestimated in the base case, while the introduction of $NH_3$ uptake leads to a lower modeled $NH_4^+$ concentration and reduced level of overestimation. For the winter, the $NH_4^+$ concentration is slightly underestimated in the base case, so the decrease of $NH_4^+$ concentration caused by the increase of $NH_3$ uptake coefficient leads to an even larger underestimation. Table 5 gives the statistics for $NO_3^-$. In general, the model over estimates the $NO_3^-$ concentration for both periods, and a poor correlation is found for the summer. The relatively poor model performance is consistent with previous CMAQ studies (Eder and Yu, 2006; Appel et al., 2008). The introduction of $NH_3$ uptake coefficient reduces the simulated $NO_3^-$ concentration significantly. The $\gamma=10^{-3}$ case leads to a mean $NO_3^-$ concentration which is much closer to the observed average than the base case in both simulated periods.

## 3.2 Air Quality Impacts

### 3.2.1 Impact on gas-phase $NH_3$ and $HNO_3$ concentrations

Figure S1 in the SI shows the time series of daily domain-averaged (averaged over 24 hours and the simulation domain) $NH_3$ for both the winter and summer, for different uptake coefficient values. In general, the $NH_3$ concentration is reduced after the introduction of the SOA-based $NH_3$ uptake process. The magnitude of the reduction is increased as the uptake coefficient increases. For the winter, the spatial-time-averaged (averaged over entire period and the simulation domain) $NH_3$ concentration for the base case is 0.44 ppb, while the value decreases to 0.43 ppb (-2.3 %) for the $\gamma=10^{-5}$ case, 0.41 ppb (-6.8 %) for the $\gamma=10^{-4}$ case and 0.31 ppb (-29.5 %) for the $\gamma=10^{-3}$ case. For the summer, the spatial-time-averaged $NH_3$ concentration for the base case is 2.30 ppb, while the value decreases to 2.10 ppb (-8.7 %) for the $\gamma=10^{-5}$ case, 1.58 ppb (-31.3 %) for the $\gamma=10^{-4}$ case and 0.76 ppb (-67.0 %) for the $\gamma=10^{-3}$ case. The impact of the uptake process is higher for the summer due to larger SOA concentrations during the summer (spatial-time-averaged 9.25 $\mu g/m^{-3}$ for the base case) than the winter (spatial-time-averaged 2.72 $\mu g/m^{-3}$ for the base case).

The spatial distribution of the impact over the simulated domain is also investigated. Figure 1 (a), (c) shows the time-averaged spatial distribution of $NH_3$ for the winter and summer base cases, while the differences between the $\gamma=10^{-3}$ case and the base case are shown in Figure 1 (b), (d). For both periods, the central valley of California is a hot spot for $NH_3$ emissions, and the region exhibits the most significant impact due to the introduction of the new $NH_3$ uptake mechanism. This is due to the intensive agricultural activities in this region including the heavy application of fertilizers (Krauter et al., 2002), and the year-round farming pattern supported by California's relatively warm climate. The hog farm industry is largely responsible for the high $NH_3$ concentration, in Carolina and north Iowa in the summer, where significant $NH_3$ loss can also be spotted in the $\gamma=10^{-3}$ case. Agriculture and wild fires also produce some hot spots of ammonia concentration in others areas, such as southern Florida in the winter and several locations in northern California and Washington state, where $NH_3$ concentrations also decreased significantly in the $\gamma=10^{-3}$ case. The spatial distribution of differences between the base case and the $\gamma=10^{-4}$





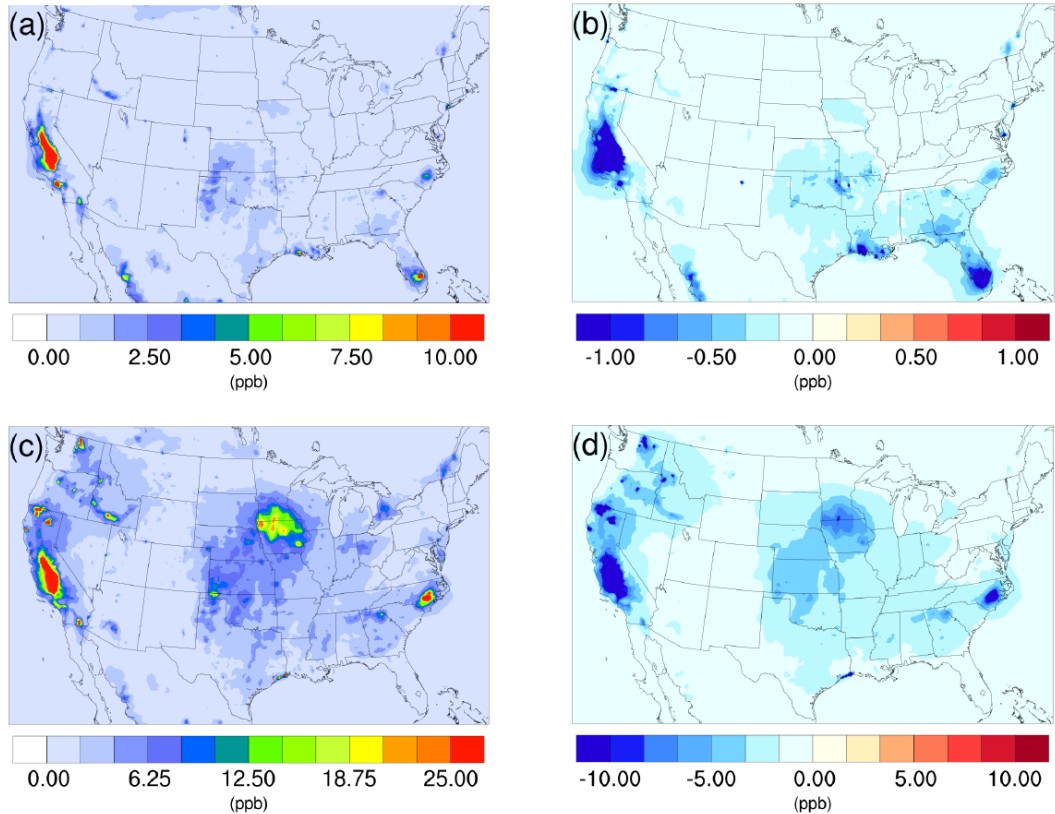

**Figure 1.** Spatial distribution of time-averaged $NH_3$ concentrations in the base case for (a) winter, and (c) summer. Spatial distribution of the difference in time-averaged $NH_3$ concentrations between the $\gamma=10^{-3}$ case and the base case for (b) winter, and (d) summer. Negative values represent decreases in concentration with respect to the base case.

and $\gamma=10^{-5}$ cases are similar to the $\gamma=10^{-3}$ only with different scales. These differences are shown in Figure S2 of supporting information.

As the condensation of $HNO_3$ into the particle phase is directly associated with $NH_3$ concentration, it is reasonable to infer that the introduction of the $NH_3$ uptake mechanism could also impact the concentration of $HNO_3$. Figure S3 in the SI shows
5 the time series of daily averaged $HNO_3$ for both the winter and summer. In contrast to $NH_3$, the integration of the $NH_3$ uptake mechanism leads to an increase in $HNO_3$ concentration, and the scale of magnitude of the increase rises as the uptake coefficient is increased, although its scale of variation is much smaller than that of $NH_3$. For the winter, the difference between the base case and the $\gamma=10^{-5}$ case is very small ($<0.2$ %), and remain insignificant for the the $\gamma=10^{-4}$ case ($\sim 1.2$ %). Only the $\gamma=10^{-3}$ case shows an significant increase in $HNO_3$ as concentrations increase by 8.5 % (the spatial-time-averaged concentration is
10 0.27 ppb for the base case and 0.30 ppb for the $\gamma=10^{-3}$ case). Similar to the $NH_3$ variation, the impact becomes larger for the summer, where the spatial-time-averaged $HNO_3$ concentration for the base case is 0.51 ppb, while the value increases by 2.0




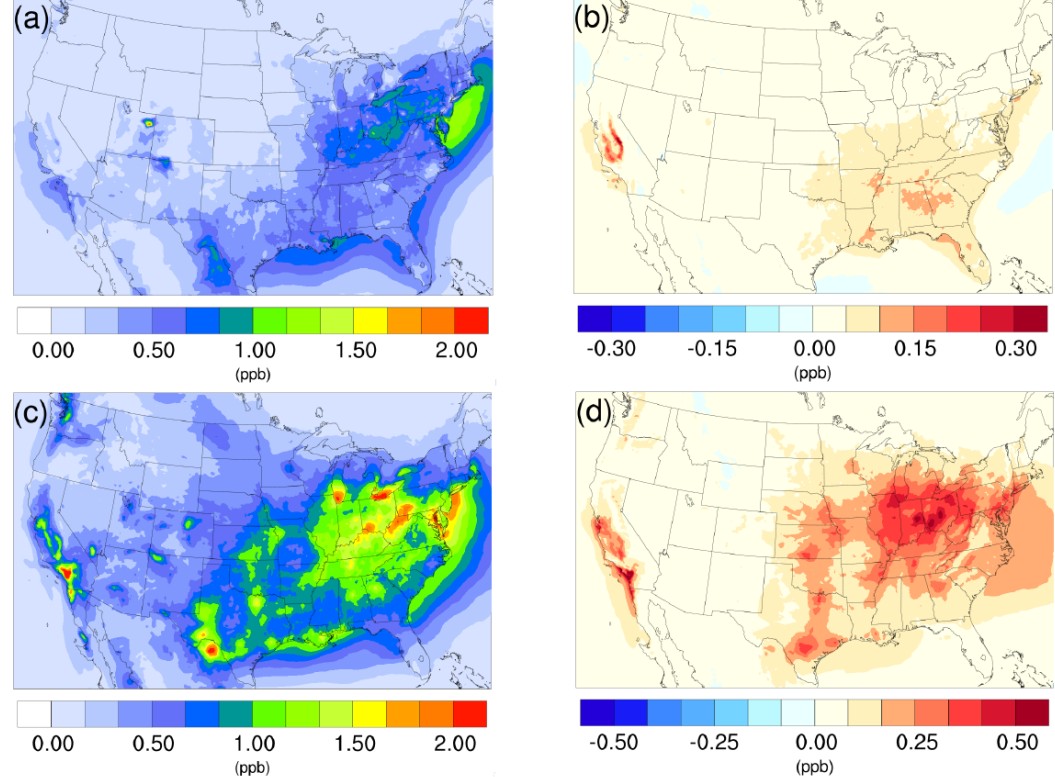

**Figure 2.** Spatial distribution of time-averaged $HNO_3$ concentrations in the base case for (a) winter, and (c) summer. Spatial distribution of the difference in time-averaged $HNO_3$ concentrations between the $\gamma=10^{-3}$ case and the base case for (b) winter, and (d) summer. Positive values represent increases in concentration with respect to the base case.

% (0.52 ppb) for the $\gamma=10^{-5}$ case, 7.8 % (0.55 ppb) for the $\gamma=10^{-4}$ case and 19.6 % (0.61 ppb) for the $\gamma=10^{-3}$ case. These increase in $HNO_3$ concentrations are due to the reduction in $NH_3$ caused by the uptake mechanism, making less $NH_3$ available for reaction with $HNO_3$ to form the particle phase $NH_4NO_3$.

The time averaged spatial distributions of $HNO_3$ for both the winter and summer base cases are presented in Figure 2 (a) and (c). The north-east region exhibits relatively high $HNO_3$ concentration for both periods, largely due to the high $NO_x$ (NO + $NO_2$) emissions from transportation activities. The introduction of $NH_3$ uptake process does not cause an obvious impact in this region for the winter, as the reduction of $NH_3$ is very small (Figure 1 (b)) due to low SOA and $NH_3$ concentrations in the base case. In contrast, the increase of $HNO_3$ becomes much more significant for this region in the summer, as the loss of $NH_3$ becomes greater due to larger $NH_3$ and SOA concentrations in the base case. The winter hot spot around northeastern Utah (Uintah Basin) could be caused by the relatively static atmospheric conditions during the winter in the valley (Lee et al., 2014), which traps $NO_x$ from local and east remote source and go under a strong nighttime reaction with $O_3$ (high $N_2O_5$ concentration is spotted in the same place). Additionally, the lack of $NH_3$ also favors the $HNO_3$ accumulation, as a result, the introduction of





$NH_3$ does not have much impact on this spot. The largest increase in $HNO_3$ concentrations in winter is found over the central valley of California, which also corresponds to the largest $NH_3$ reduction (Figure 1 (b)). For the summer, the largest impact occurs over the hot spot of southern California, where strong traffic emissions of $NO_x$ and active photo-chemistry provide strong $HNO_3$ source. The significant reduction of $NH_3$ concentration from the south central valley could reduce the potential

sink of $HNO_3$ into particle-phase and leave more $HNO_3$ in the gas-phase. The spatial distribution of differences between the base case and the $\gamma=10^{-4}$ and $\gamma=10^{-5}$ cases are similar to the $\gamma=10^{-3}$ only with different scales, and they can be found in the SI (Figure S4).

### 3.2.2   Impact on inorganic PM

One of the effects of the gas-phase $NH_3$ reduction due to the inclusion of SOA-based $NH_3$ uptake mechanism would be the

decrease of $NH_4^+$ concentration in the particle phase, as all $NH_4^+$ originates from gas phase $NH_3$. Figure S5 in the SI shows the time-spatial evolution of daily averaged $NH_4^+$ for the winter and the summer. In general, the introduction of $NH_3$ uptake in the model causes a decrease in particle phase $NH_4^+$ concentration, and the impact is more significant for the summer than the winter. For summer case, the average decrease in $NH_4^+$ is 1.8 % for $\gamma=10^{-5}$, 10.7 % for $\gamma=10^{-4}$ and 28.2 % for $\gamma=10^{-3}$; for winter case, the averaged decrease is 0.2 % for $\gamma=10^{-5}$, 2.3 % for $\gamma=10^{-4}$ and 13.2 % for $\gamma=10^{-3}$. Such behavior corresponds

well to the level of $NH_3$ reduction in Figure S1, and is caused by the higher SOA concentrations during the summer.

The time-averaged spatial distributions of the $NH_4^+$ concentration for both the winter and summer base case are shown on Figure 3 (a) and (c). Most of the $NH_4^+$ is concentrated over the eastern part of the US, as a result of high $NH_3$ concentrations (see Figure 1) in this region combined with the abundance of $NH_3$ neutralizers (e.g., $HNO_3$ and $H_2SO_4$). Another hot spot is the Central Valley of California and the South Coast Air Basin of California, resulting from high $NH_3$ emissions from the intensive

agriculture (Figure 1). In presence of both $HNO_3$ and $H_2SO_4$, $NH_3$ is first neutralized by $H_2SO_4$ to form either $(NH_4)_2SO_4$ or $NH_4HSO_4$ in the particle phase, while the rest of the $NH_3$ reacts with $HNO_3$ and forms particle phase $NH_4NO_3$. The association form of $NH_4^+$ could be investigated by comparing the spatial distribution of the $NO_3^-$ concentration for corresponding period in Figure 4 (a) (c) and the $SO_4^{2+}$ in Figure 5 (a) (b). For the winter, the $H_2SO_4$ concentration is insufficient to neutralize all the $NH_3$ for the mid-east region, so more $NO_3^-$ is involved in the $NH_3$ neutralization, and there are more nitrate particles than

sulfate particles. For the summer, as the sulfate concentration almost doubles over the mid-east US compares to the winter, most of the $NH_3$ is neutralized by $H_2SO_4$. This causes a absence of $NO_3^-$ above this region, and only appears on the surrounding region where sulfate concentration is low. For the West Coast and the Central Valley of California, the enriched $NH_4^+$ mostly exists in the form of $NH_4NO_3$, as the sulfate concentration is low in this region for both periods. Figure 3 (b) and (d) present the spatial distribution of the difference in $NH_4^+$ concentration between the $\gamma=10^{-3}$ case and the base case, which is highly

correlated with the $NH_3$ variation map (Figure 1). The reduction in $NH_3$ due to the SOA uptake, directly impacts the available $NH_3$ that could be condensed into the particle phase, and reduces the $NH_4^+$ concentration consequently. The spatial distribution of differences between the base case and the $\gamma=10^{-4}$ and $\gamma=10^{-5}$ cases is similar to the $\gamma=10^{-3}$ only with different scales, as shown in Figure S6 in the SI.





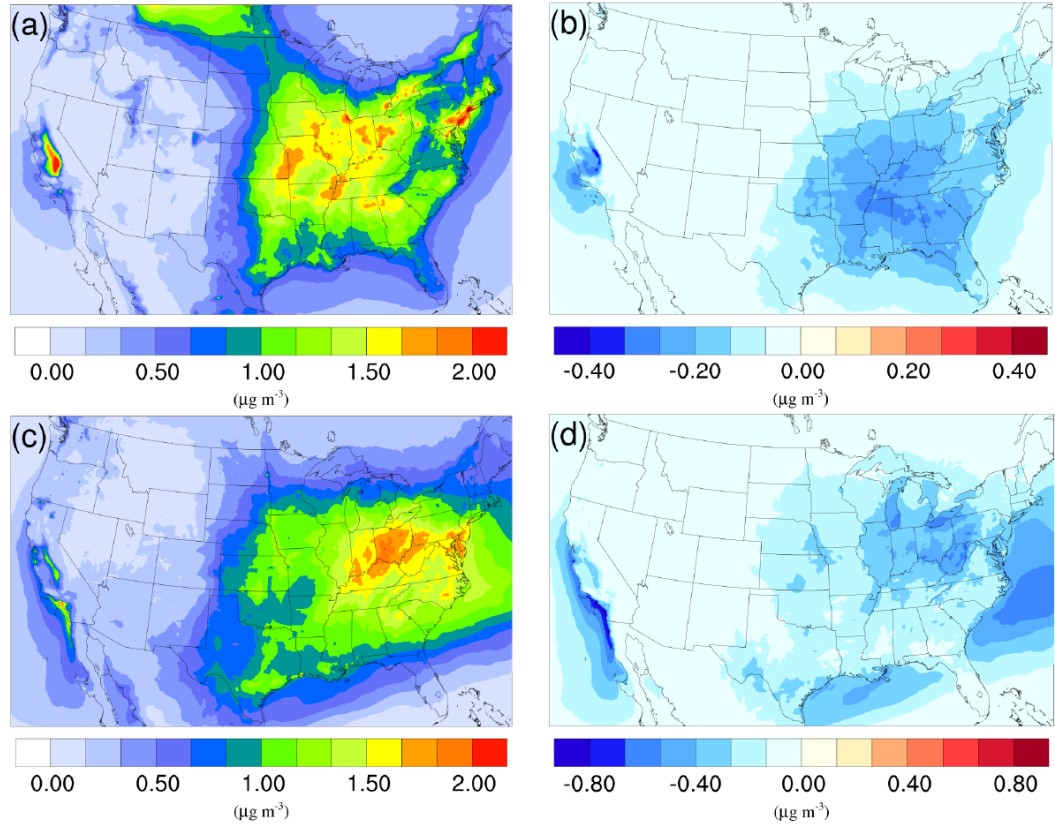

**Figure 3.** Spatial distribution of time-averaged $NH_4^+$ concentrations in the base case for (a) winter, and (c) summer. Spatial distribution of the difference in time-averaged $NH_4^+$ concentrations between the $\gamma=10^{-3}$ case and the base case for (b) winter, and (d) summer. Negative values represent decreases in concentration with respect to the base case.

The concentration of $NO_3^-$ also changes as a result of adding the $NH_3$ uptake mechanism. Figure S7 in the SI shows the variation in daily-spatial averaged $NO_3^-$ concentration under different scenarios for both the winter and summer. Overall, adding the $NH_3$ uptake mechanism leads to a decrease in $NO_3^-$ concentrations for both periods. Similar to $NH_4^+$, the impact is more significant for the summer than the winter. The average reductions for the winter are 0.2 % for $\gamma=10^{-5}$, 1.9 % for
5   $\gamma=10^{-4}$ and 10.9 % for $\gamma=10^{-3}$. For the summer, the average reductions are 1.9 % for $\gamma=10^{-5}$, 10.6 % for $\gamma=10^{-4}$ and 24.3 % for $\gamma=10^{-3}$. Such variations are similar to those of $NH_4^+$, where the $\gamma=10^{-5}$ case in the summer has similar reductions to $\gamma=10^{-4}$ case in the winter. And the magnitude of the difference is also close to the difference in $NH_4^+$, indicating almost all the $NH_4^+$ reduction is from $NH_4NO_3$.

    The spatial distributions of the $NO_3^-$ variation due to the addition of the $NH_3$ uptake mechanism ($\gamma=10^{-3}$) are presented on
10   Figure 4 (b) (d) for the winter and summer. By comparing with the base cases (see Figure 4 (a) (c)), it is clear that most of the $NO_3^-$ reduction occurs over regions with high $NO_3^-$ concentration, such as the Central Valley of California, the South Coast





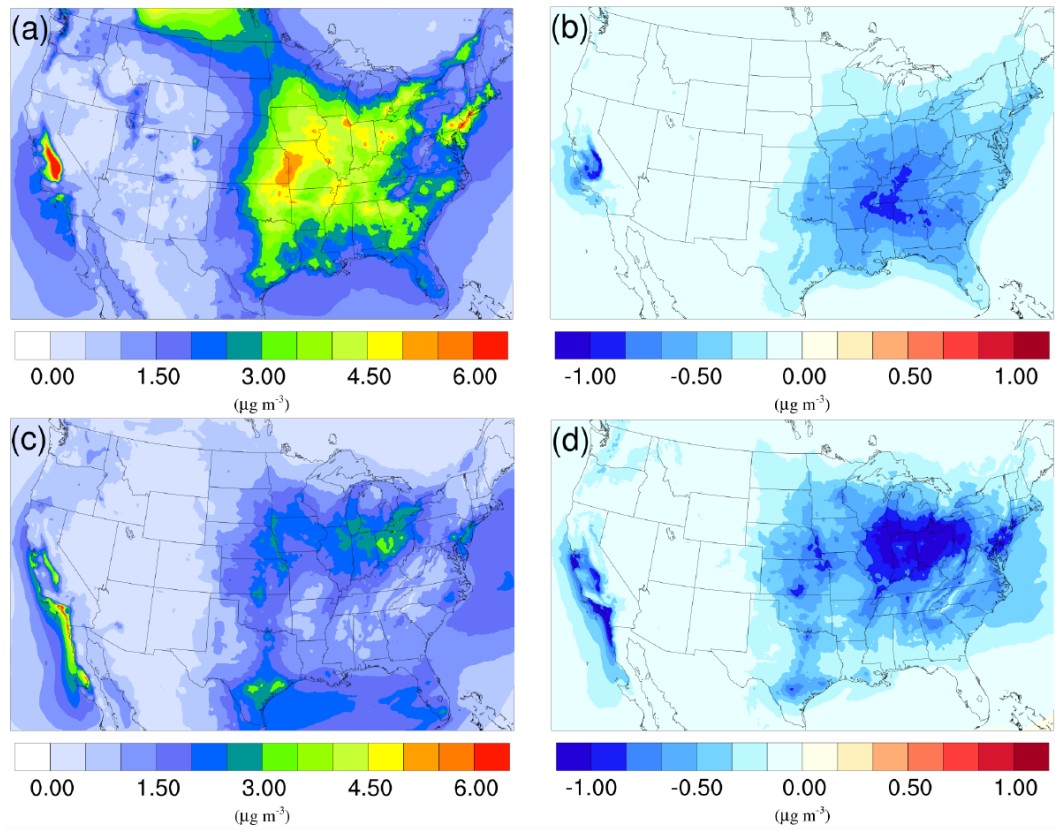

**Figure 4.** Spatial distribution of time-averaged $NO_3^-$ concentrations in the base case for (a) winter, and (c) summer. Spatial distribution of the difference in time-averaged $NO_3^-$ concentrations between the $\gamma=10^{-3}$ case and the base case for (b) winter, and (d) summer. Negative values represent decreases in concentration with respect to the base case.

Air Basin of California and vast regions over the mid-east US. One exception is the high $NO_3^-$ region over Canada on the north edge of Montana and North Dakota during the winter. Neither $NH_4^+$ concentration nor $NO_3^-$ concentration changes much for that region, mostly because the SOA concentration is extremely low for that region (see Figure 6 (a)), so almost no $NH_3$ is lost due to the SOA uptake. The same thing also occurs in south Florida during the summer. The spatial distribution of differences
5   between the base case and the $\gamma=10^{-4}$ and $\gamma=10^{-5}$ cases is similar to the $\gamma=10^{-3}$ only with different scales, shown in Figure S8 of the SI.

### 3.2.3   Impact on organic PM

Figure 6 (a), (c) shows the time-averaged spatial distribution of SOA for the winter and summer base case. For both seasons, high SOA concentrations are found over the southeastern US due to high vegetation coverage in this region, while hot spots
10   in the northwestern region are caused by wide fire events. The averaged SOA concentration is more than 3 times higher in the





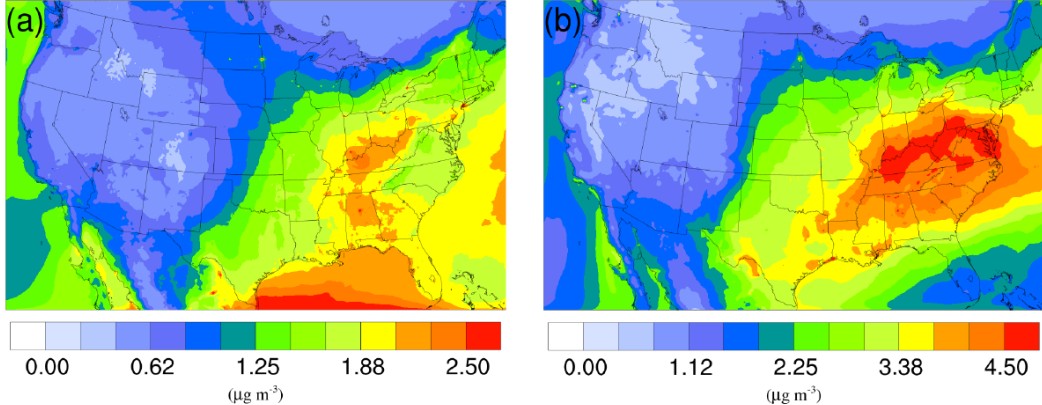

**Figure 5.** Spatial distribution of time-averaged $SO_4^{2-}$ concentrations in the base case for (a) winter, and (b) summer.

summer case (9.25 $\mu$g m$^{-3}$) than in the winter (2.72 $\mu$g m$^{-3}$), largely due to the much higher biogenic SOA concentrations (4.43 $\mu$g m$^{-3}$ summer vs. 0.22 $\mu$g m$^{-3}$ winter) resulting from elevated biogenic emissions in the warm season.

 As demonstrated in Figure 6 (b), (d), implementing of the NH$_3$ uptake mechanism has a significant impact on the SOA concentrations during the summer, but has almost no impact on SOA for the winter. Almost the entire increase in SOA con-

5 centrations in the summer is due to the mass change in biogenic SOA (BIOSOA) (see Figure 7 (a) and 6 (d), their average concentrations for the base case are in the SI Figure S9). Further investigation reveals that the majority of the increase ($\sim$ 80%) is caused by the nonvolatile AISO3 species (7 (b)), which is the isoprene epoxydiols (IEPOX) derived SOA through the acid-catalyzed ring-opening reactions (Pye et al., 2013). This increase in AISO3 is caused by the increase of aerosol aqueous phase acidity due to the reduction in NH$_4^+$ after adding the NH$_3$ uptake mechanism. This increase in particle acidity corre-

10 sponds well with the sensitivity study between NH$_3$, SO$_4^{2-}$ and particle pH presented in Figure 2 of Weber et al. (2016), where particle pH is found to be more sensitive to NH$_3$ concentrations than to SO$_4^{2-}$ concentrations. Figure 7 (c) shows a large drop in pH value ($\sim$0.9 - 2.3) (pH change for other scenarios are shown in SI Figure S10) in the southeast region where the increase of the AISO3 is most significant and there is a simultaneous decrease in IEPOX concentrations (Figure 7 (d)). The largest pH variation appears over the northwest region. However, there is no observable impact on SOA concentrations due to the

15 extremely low concentration of both isoprene and IEPOX (see Figure 7 (e) and (f)) in this area. Moreover, the reduction in NH$_4^+$ concentrations also increases the ratio of SO$_4^{2-}$/HSO$_4^-$, where SO$_4^{2-}$ can acts as a nucleophile and promote the IEPOX uptake process. This also contributes to the increase of AISO3 in the $\gamma$=10$^{-3}$ case.

 Figure 8 shows the time evolution of daily-spatial averaged H$^+$, IEPOX and AISO3 for both the winter and summer. Although the average H$^+$ concentration in the base case is similar between two periods, the variation is much smaller for the

20 winter largely due to the lower SO$_4^{2-}$ concentrations in the winter which restraints the acidity variation level. Additionally, lower SOA concentrations in winter also reduces the magnitude of NH$_4^+$ variation. As a result, addition of the NH$_3$ uptake mechanism does not have large impact on the AISO3 concentration for most of the simulation (except for the last several





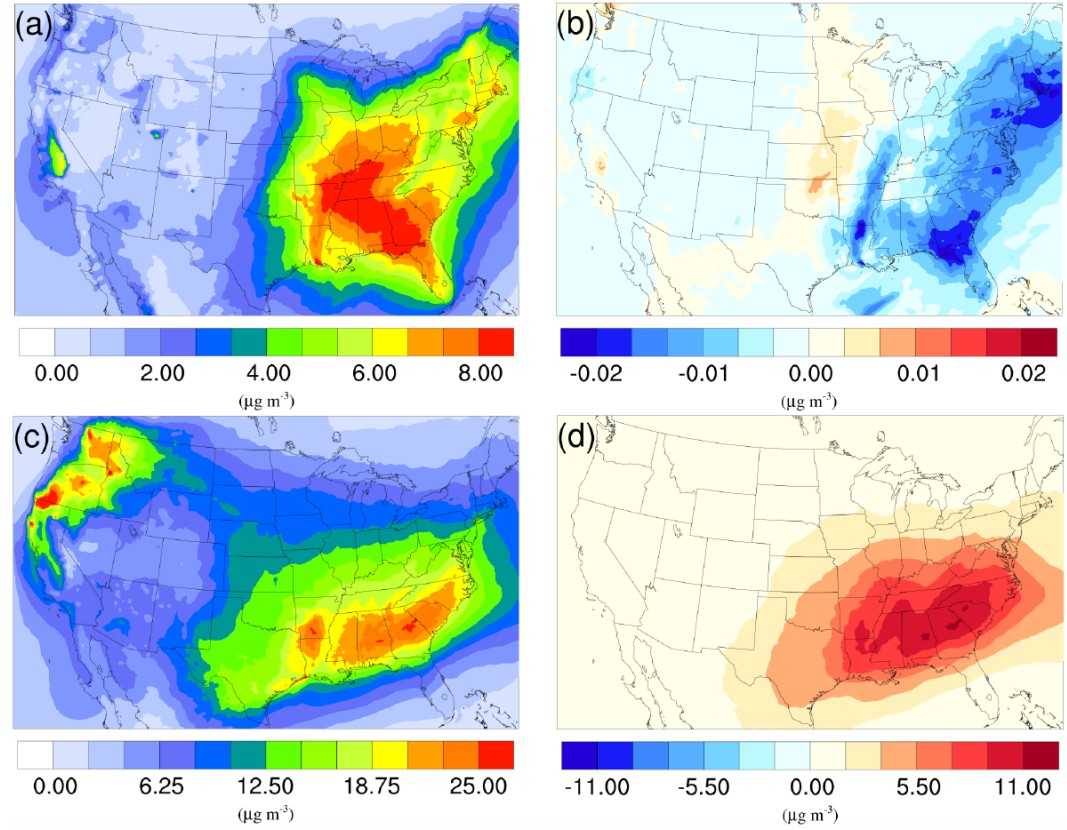

**Figure 6.** Spatial distribution of time-averaged SOA concentrations in the base case for (a) winter, and (c) summer. Spatial distribution of the difference in time-averaged SOA concentrations between the $\gamma=10^{-3}$ case and the base case for (b) winter, and (d) summer. Positive values represent increases in concentration with respect to the base case, and negative values represent decreases in concentration with respect to the base case.

days). On the contrary, the summer shows a significant increase in $H^+$ concentrations as the $NH_3$ uptake coefficient increases, while the concentration of IEPOX decrease. And the increase of AISO3 concentration is remarkable, with more than ten times growth on average between the $\gamma=10^{-3}$ case (1875.2 ng m$^{-3}$) and the base case (181.75 ng m$^{-3}$). The amount of growth on AISO3 seems exponential with different value of the $NH_3$ uptake coefficient ($\gamma=10^{-5}$: 16.2%; $\gamma=10^{-4}$: 171.9%; $\gamma=10^{-3}$:

5    931.6%).

Beside the isoprene epoxydiols pathway, other biogenic SOA species contribute the rest of the SOA changes ( 20%), including other SOA species derived from isoprene (AISO1and AISO2), from monoterpenes (ATRP1 and ATRP2), from sesquiter-penes (ASQT), and AOLGB which represents the aged nonvolatile SOA origin from AISO1, AISO2, ATRP1, ATRP2 and ASQT. The common point with those SOAs (AISO1, AISO2, ATRP1, ATRP2 and ASQT) are that they all have a pathway to

10   be formed through the oxidation between $NO_3$ and their gas precursors. One possible explanation could be that the introduction

**Figure 7.** Spatial distribution of the difference in time-averaged (a) biogenic SOA concentrations, (b) isoprene epoxydiols (IEPOX) derived SOA concentrations, (c) particle acidity (pH), and (d) isoprene epoxydiols concentrations between the $\gamma=10^{-3}$ case and the base case during the summer. Spatial distribution of time-averaged (e) isoprene, and (f) isoprene epoxydiols concentration in the base case during the summer.

of $NH_3$ uptake leads to an increase of gas phase $HNO_3$, which could shift the reaction balances between $NO_3$ and $HNO_3$ and leave more $NO_3$ available for SOA oxidation.



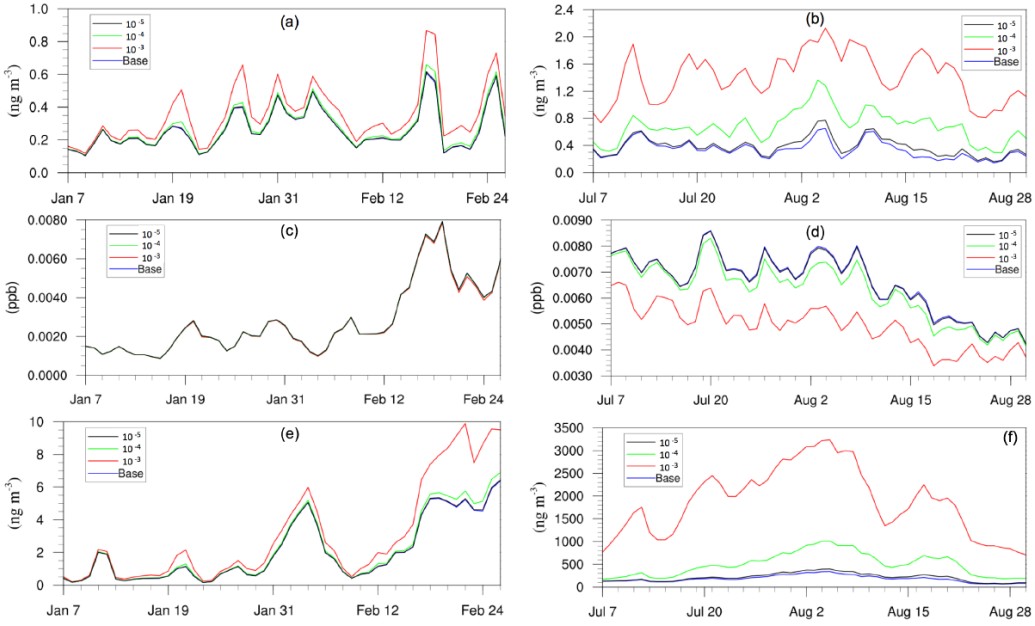

**Figure 8.** Daily, spatially-averaged concentrations of (a) particle phase $H^+$ in winter, (b) particle phase $H^+$ in summer, (c) isoprene epoxydiols in winter, (d) isoprene epoxydiols in summer, (e) isoprene epoxydiol derived SOA in winter, and (f) isoprene epoxydiol derived SOA in summer.

### 3.2.4 Impact on total PM

Figure S11 in the SI presents the time evolution of daily-averaged concentrations of $PM_{2.5}$ and $PM_{10}$ in different scenarios during both periods. First, both the pattern and level of impact caused by the $NH_3$ uptake mechanism is similar for $PM_{2.5}$ and $PM_{10}$, which indicates that most of the mass change due to this process occurs on fine particles. Secondly, the level of impact on both $PM_{2.5}$ and $PM_{10}$ is much more significant over the summer than the winter, which is consistent with previous analysis of individual species. Third, opposite impact patterns are found between the winter and summer. The inclusion of $NH_3$ uptake mechanism leads to a decrease in the total PM mass for the winter, that is caused by the reduction of inorganic $NH_4^+$ and $NO_3^-$ due to the decrease of $NH_3$ concentration, as detailed in section 3.2.2. On the contrary, PM concentrations during the summer increases after adding the $NH_3$ uptake mechanism. Although the concentration of inorganic species still decreases during the summer, the increase in biogenic SOA concentration, as detailed in section 3.2.3, outpaces the decrease caused by inorganic species and leads to an overall increase in total PM mass for the summer. For the winter, the average $PM_{2.5}$ concentration reduction is 0.07% for the $\gamma=10^{-5}$ case, 0.59% for the $\gamma=10^{-4}$ case and 3.39% for the $\gamma=10^{-3}$ case. For the summer, the average $PM_{2.5}$ concentration increase is 0.14% for the $\gamma=10^{-5}$ case, 2.05% for the $\gamma=10^{-4}$ case and 12.38% for the $\gamma=10^{-3}$ case.





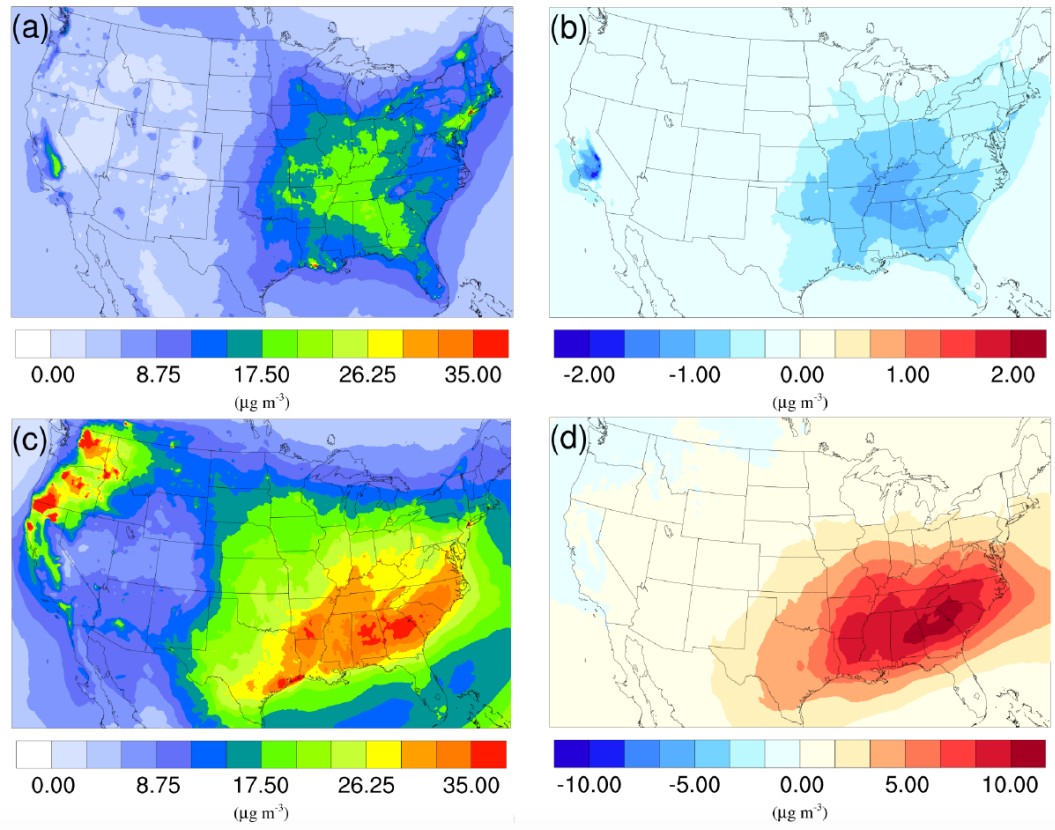

**Figure 9.** Spatial distribution of time-averaged PM$_{2.5}$ concentrations in the base case for (a) winter, and (c) summer. Spatial distribution of the difference in time-averaged PM$_{2.5}$ concentrations between the $\gamma=10^{-3}$ case and the base case for (b) winter, and (d) summer. Positive values represent increases in concentration with respect to the base case, and negative values represent decreases in concentration with respect to the base case.

The spatial distribution of time averaged PM$_{2.5}$ concentration for the winter and summer is presented in Figure 9 (a) and (c) respectively. Most of the high PM$_{2.5}$ concentration happens over the mid-east US during the winter, with additional hot spots over the Central Valley of California, resulting in an overall average of 7.47 $\mu g/m^3$. PM$_{2.5}$ concentrations are highly correlated with the population density map of the US, indicating a dominant anthropogenic origin. The relatively low fraction of biogenic

5 SOA in winter also supports this point (Figure 10 (a)). The model predicts a much higher PM$_{2.5}$ concentration for the summer, with an average concentration of 16.17 $\mu g/m^3$. The hot spots observed over the northwest of the country and coastal area over southeast Texas are caused by wild fire events. In general, high PM$_{2.5}$ concentration over the southeast of the US, where high fractions of biogenic SOA are found (Figure 10) (b). This could be a result of both high average temperatures during the summer and high vegetation density in that region. Figure 9 (b) shows the variation in PM$_{2.5}$ concentrations between the

10 $\gamma=10^{-3}$ case and the base case for the winter. An overall reduction can be observed from the map, with the highest reduction





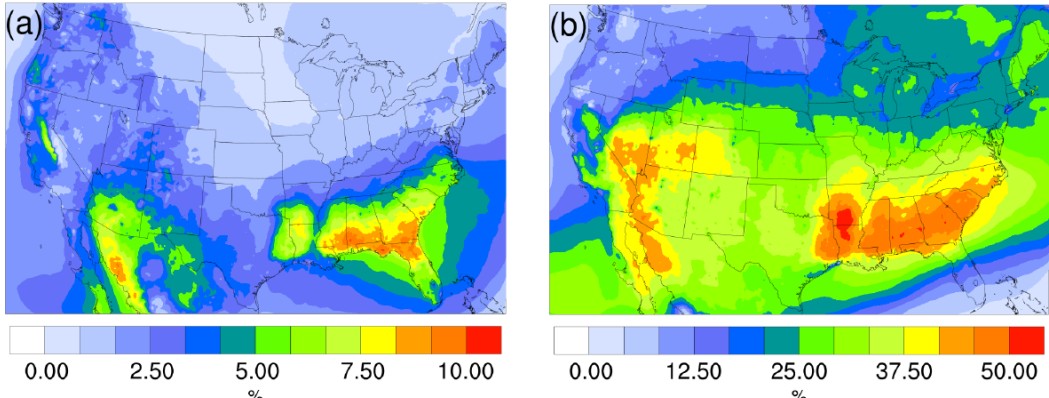

**Figure 10.** Spatial distribution of time-averaged biogenic SOA fraction of total $PM_{2.5}$ for (a) the winter, and (b) summer.

around the Central Valley of California and a smaller reduction over the vast mid-east region. This is mostly caused by the decrease of $NH_4NO_3$ due to the reduction of gas-phase $NH_3$ concentrations as discussed in section 3.2.2. For the summer, although the decrease still appears over the northwest of the country, the prominent feature becomes a significant increase in $PM_{2.5}$ concentrations over the southeast region. This is due to the increase in biogenic SOA resulting from the enhanced

acid-catalyzed ring-opening reactions as detailed in section 3.2.3.

## 4   Conclusions

In this study, the potential air quality impacts of the heterogeneous uptake of $NH_3$ by SOA is investigated with the CMAQ model. Simulations over the continental US are performed for the winter and summer seasons of 2011 with a range of $NH_3$ uptake coefficients reported in the literature. First, the simulation results for the two base case simulations are compared with

observation data from different monitoring networks, and statistics show an overall good model performance for most of the criteria. The inclusion of the SOA-based $NH_3$ uptake mechanism has a significant impact on the statistics of $NH_3$, $NH_4^+$, $NO_3^-$, but does not affect $O_3$ and $SO_4^{2+}$. The overestimation of $NH_3$ and $NH_4^+$ for the summer is reduced by this new mechanism. Moreover, the prediction of $NO_3^-$ is improved by this mechanism, given that the overestimation of $NO_3^-$ concentration gradually subsides as the uptake coefficient increases.

The comparison between different uptake coefficient scenarios and the base case allows a more detailed understanding of the impact of this mechanism on both gas phase and particle phase species. Simulation results indicate a significant reduction in gas-phase $NH_3$ possibly due to the uptake by SOA, and such reduction increases dramatically as the uptake coefficient increases. The highest spatially-averaged reduction in gas-phase $NH_3$ is 31.3 % in the winter and 67.0 % in the summer. This analysis is based on a range of uptake coefficient that span those reported in the literature. However, the actual value for each

individual SOA could be lower or higher than the fixed uptake coefficient used in this study, although the magnitude of the



impact still indicates the importance of including this process in air quality models. The seasonal differences are obvious as the impact is much more significant in the summer than in the winter, due to much higher $NH_3$ and SOA concentration in the summer. The concentration of gas-phase $HNO_3$ is also impacted by this new mechanism. As the $NH_3$ concentration drops, less $HNO_3$ is neutralized by $NH_3$, resulting in an overall increase in $HNO_3$ concentration. Such increases can be as high as 8.5% in

the winter and 19.6% in the summer for the largest uptake coefficient. Geographically, the biggest reduction in $NH_3$ happens in the Central Valley of California during both seasons, the same location as the biggest increase in $HNO_3$ in the winter. While for the summer, $HNO_3$ increases more dramatically over the South Coast Air Basin of California and the northeast region of the country.

      PM concentrations are found to decrease during the winter period, largely due to the reduction in ammonium nitrate for-

mation causes by the decrease in gas-phase ammonia. The largest uptake scenario leads to a 13.2% reduction of $NH_4^+$, 10.6% reduction of $NO_3^-$ and 3.4% reduction of $PM_{2.5}$ in the winter. The most significant reduction also happens over the Central Valley of California region with a highest $PM_{2.5}$ drop of 2.0 $\mu g/m^3$. On the other hand, PM concentrations are found to increase during the summer due to the increase in biogenic SOA production resulting from the enhanced acid-catalyzed ring-opening reactions. Although the reduction in ammonium nitrate is even larger in magnitude during the summer (28.2% reduction in

$NH_4^+$, 24.3% reduction in $NO_3^-$) than the winter, the dramatic increase in biogenic SOA outpaced the decrease caused by ammonium nitrate to result in an overall increase in total PM (12.4% increase in $PM_{2.5}$). Most of the biogenic SOA increases occur over the southeast region of the US, where high vegetation density is located. The average increase in biogenic SOA is 0.9% for $\gamma=10^{-5}$, 9.2% for $\gamma=10^{-4}$ and 49.0% for $\gamma=10^{-3}$. For the species (AISO3) that is responsible for most of the increase, the $\gamma=10^{-3}$ case leads to a 10-fold increase in concentration compared to the base case.

Results of this study show that the chemical uptake of $NH_3$ by SOA can have significant impact on the model-predicted concentration of important atmospheric pollutants, including $NH_3$, $HNO_3$, $NH_4^+$, $NO_3^-$ and biogenic SOA. The impact on the total PM has a distinct pattern on different seasons. Future laboratory studies should be conducted to identify the nature of the chemical interaction between $NH_3$ and SOA species to provide more accurate model representation of the uptake process. For example, single particle measurements conducted by Neuman et al. (2003) showed that organic aerosols also contributed

to increases in fine-particle mass in regions with high $NH_3$ emissions rates, suggesting that $NH_3$ uptake can increase organic aerosol mass concentrations directly. Current air quality models only include one pathway for the acid-catalyzed SOA generation (based high $NO_x$ assumption (Pye et al., 2013)), and a more detailed representation of other acid-catalyzed pathways could lead to even larger impact on the SOA concentration.

*Code and data availability.* Simulation result data sets are available upon request as they are too big to upload online (812 Gigabyte). The

original CMAQ (version 5.2) code for the base case simulation is available on the CMAS website: https://www.cmascenter.org/cmaq/. The updated CMAQ code including the $NH_3$ uptake mechanism is available under the following link: http://albeniz.eng.uci.edu/software/CMAQv5.2_withNH3Uptake.zip. CMAQ have a GNU (General Public License). The user can redistribute them and/or modify them under the terms of the GNU General Public License as published by the Free Software Foundation.



*Competing interests.* The authors declare that they have no conflict of interest.

*Acknowledgements.* This publication was developed under Assistance Agreement No. EPA 83588101 awarded by the U.S. Environmental Protection Agency to the Regents of the University of California. It has not been formally reviewed by EPA. The views expressed in this document are solely those of the authors and do not necessarily reflect those of the Agency. EPA does not endorse any products or commercial

5   services mentioned in this publication. We also express our gratitude for UCI HPC assistance and especially Dr. Harry Mangalam and Garr Updegraff for their generous support, and the UCI Research Computing group especially Allen Schiano and Dana Roode.



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
