# Peer review of "Modeling reactive ammonia uptake by secondary organic aerosol in CMAQ: application to continental US"

_Atmospheric Chemistry and Physics, 2017_

## Referee Comment (RC1) · Anonymous Referee #2 · 8 Nov 2017

General Comments

This work describes the effects of including NH3 uptake onto secondary organic aerosol (SOA) in the CMAQ model. A range of NH3 uptake coefficients are used, taken from a recent lab study. The authors find that the addition of this reaction significantly lowers gas phase NH3 concentrations, especially when higher uptake coefficients (10ˆ-3) are used. Lower NH3 concentrations then cause less ammonium nitrate to be formed, and higher HNO3 gas phase concentrations. During the winter, this effect is dominant, and PM2.5 and PM10 levels decline. During the summer, lower NH3 concentrations increase the aerosol acidity significantly (pH drops by ∼2),

which then triggers SOA production by acid-catalyzed pathways, especially by IEPOX species. This work will be of interest not only to modelers but also to those who study the chemistry of either inorganic or organic aerosol species in the lab and in the field.

The primary limitation of the study is that once NH3 is taken up by SOA in the model, it disappears. As described in the introduction, NH3 can be taken up into SOA by either neutralizing organic acids (producing ammonium salts) or by reacting with aldehyde species to produce NOC (nitrogen-containing organic carbon species, such as imines and imidazoles), most of which are still quite basic and could react with inorganic acids. The relative importance of these two competing reactions is not known, but this study neglects both options. The result is the counterintuitive conclusion that including NH3 uptake to aerosol particles in the model reduces both NH3 (gas) and NH4+ (aerosol) concentrations, while also increasing HNO3 (gas) and decreasing NO3- (aerosol) concentrations. To a great extent, NH3 uptake to SOA must either produce NH4+ in the aerosol particle (by neutralizing an organic acid) or produce basic NOC species that can still neutralize HNO3. NH3 uptake that generates neither of these products, as assumed in this manuscript, does not appear to be a viable option. While the state of knowledge of this chemistry is not quantitative enough to nail this down, and the authors allude to this in the last paragraph in the paper, these issues should be discussed more vigorously in the manuscript.

Specific Comments

It would be helpful to mention whether aerosol in the model are externally or internally mixed.

p. 11 line 8: This sentence implies that both California's central valley and the South Coast Air Basin have high NH3 emissions from intensive agriculture. Is this really true in the latter case?

p. 11 line 9: Where do organic acids fit in the order of NH3 neutralization with H2SO4 and HNO3? If NH3 uptake to SOA results in neutralization of organic acids, does this

affect any of the manuscript's conclusions about HNO3 (g) concentrations increasing and nitrate concentrations decreasing in response to NH3 uptake?

p. 12 line 4: The prediction of almost no nitrate in summer aerosol over the southeast U.S., due to sulfuric acid neutralizing all of the available NH3, should be testable against regional PM observations. Is the prediction consistent with this dataset?

p. 12 line 9: This sentence is an example of the strange reasoning caused by the lack of a product formed by NH3 uptake in the model. "The reduction in NH3 due to the SOA uptake, directly impacts the available NH3 that could be condensed into the particle phase, and reduces the NH4+ concentration considerably."

Technical Corrections

p. 3 line 31: the phrase in parentheses does not make sense.

p. 9 line 10: the sentence with the phrase "east remote source and go under. . ." does not make sense. In the following sentence, when the authors write "the introduction of NH3 does not have much impact on this spot" do they mean "the addition of NH3 uptake to the model does not have much impact at this location"?

p. 11 line 10: the meaning of "association form of NH4+" is unclear.

p. 13 line 21: "wide" should be "widespread"

p. 16 line 14: The growth of AISO3 with respect to the uptake coefficient is linear, not exponential, since the uptake coefficients were varied exponentially.

p. 20 line 16: the meaning of the parenthetical phrase "based high NOx assumption" is unclear.

---

## Referee Comment (RC2) · Anonymous Referee #1 · 6 Jan 2018

This manuscript presents a model sensitivity analysis to quantify the effects of NH3 uptake by SOA on surface PM2.5 in the US. While this is the first modeling study I know of that simulates the regional effect of this reaction, the effect is parametrized simply as a sink for NH3, not a source for SOA, and thus the air quality impacts resulting from this are all related to changes in the inorganic PM species and aerosol acidity. The aerosol acidity change in turn drives some SOA changes resulting from the acid-catalyzed SOA formation pathways. The paper is generally well-written. I recommend it to publish in ACP after the following weakness/questions are addressed.

Major Comments:

[Figure]

1. Pg 4, line 20-29: the discussion of lab studies would indicate the parameterization used in the manuscript is oversimplified and may not fully represent the lab experiments. First, lab experiments show that only about 10% of SOA molecules can react with NH3 to form nitrogen-containing organic compounds. Second, the Liu et al. (2015) study, which the parameterization is largely based on, reported only a few SOAs can uptake NH3. Despite these, the manuscript assumes all SOA uptakes NH3. Although they chose to lower the uptake coefficient to compensate for the fact that not all SOA uptakes ammonia, given the spatiotemporal variability of SOA sources, a uniformly-applied lower uptake coefficient to all the SOA species would not have the same effect as that of applying a higher uptake coefficient selectively to several SOAs. Since the CMAQ model can explicitly simulate SOA species by origin (e.g. isoprene vs. terpene SOA) and by oxidation pathways, would it make more sense to parameterize the uptake only to the few SOA species that lab experiments have shown to have such ability and use the lab derived uptake coefficient directly?

2. Model evaluation: this section presents just general PM evaluation and does not have a clear focus on evaluating the parameterization scheme developed. At a minimum, the sites where the model evaluation is based on should be labeled on the model concentration maps from Figure 1 – 6 with the corresponding model biases. The way they are presently listed in Tables is not illustrative and does not help the readers understand the simulation results in the context of model biases. The concentration maps show large spatial variability of the NH3 uptake effect on different aerosol components (e.g. over Southeast US, Central Great Plains), but they are all based on model simulations without observational backup. If there are some observational sites located within those regions where large model sensitivity is found, more discussions of the model bias or improvement after the parameterization scheme should be given to them.

3. Section 3.2.3: the impact of the NH3 uptake on organic PM is large, which apparently contradicts with the earlier claim (pg 4, line 10-15) that the NH3 uptake does not create SOA mass. Later on the authors explained that it was due to changes in

aerosol acidity, which in turn drives SOA changes from the acid-catalyzed formation pathways. To avoid confusion, I suggest the authors first present the acidity changes (maybe make it a separate sub-section) and then present the acidity-induced SOA changes. In the SOA section, state upfront that the SOA changes are not caused by the parameterization creating more SOA mass by itself.

Technical Comments:

1. Equation 1: is the aerosol surface area (Sj) dry or wet area? I think it should be wet area, i.e. considering hygroscopic growth of aerosols under ambient RH conditions. If wet area, how does the model treat SOA hygroscopic growth?

2. Pg 4, Line 10: "an SOA compounds" should be "an SOA compound"

---

## Author Comment (AC1) · 1 Feb 2018

**Interactive Comment reply to Anonymous Referee #1**

Shupeng Zhu[1], Jeremy R. Horne[1], Julia Montoya-Aguilera[2], Mallory L. Hinks[2], Sergey A. Nizkorodov[2], and Donald Dabdub[1]

[1]Computational Environmental Sciences Laboratory, Department of Mechanical & Aerospace Engineering, University of California, Irvine, Irvine, CA, 92697-3975, USA

[2]Department of Chemistry, University of California, Irvine, Irvine, CA, 92697-3975, USA

The authors thank the referee for providing a thorough review and agree that some changes and clarification will improve the manuscript. We would propose to make the revisions outlined below for submission to *Atmospheric Chemistry and Physics*. Each item starts with the reviewer's comment in bold followed by our response in plain text and blue color.  Page numbers refer to the revised version of the paper. The updated manuscript and supporting information along with the marked-up version of the manuscript are all attached at the end of this response.

**Major Comments:**

1. **Pg 4, line 20-29: the discussion of lab studies would indicate the parameterization used in the manuscript is oversimplified and may not fully represent the lab experiments. First, lab experiments show that only about 10% of SOA molecules can react with NH3 to form nitrogen-containing organic compounds. Second, the Liu et al. (2015) study, which the parameterization is largely based on, reported only a few SOAs can uptake NH3. Despite these, the manuscript assumes all SOA uptakes NH3. Although they chose to lower the uptake coefficient to compensate for the fact that not all SOA uptakes ammonia, given the spatiotemporal variability of SOA sources, a uniformly applied lower uptake coefficient to all the SOA species would not have the same effect as that of applying a higher uptake coefficient selectively to several SOAs. Since the CMAQ model can explicitly simulate SOA species by origin (e.g. isoprene vs. terpene SOA) and by oxidation pathways, would it make more sense to parameterize the uptake only to the few SOA species that lab experiments have shown to have such ability and use the lab derived uptake coefficient directly?**

    We acknowledge the reviewer's concern on the simplified nature of the parameterization used in this study. The use of uniform $NH_3$ uptake for all types of SOA is the best assumption we can make given the current state of knowledge, as the current laboratory data are not detailed enough to provide specific uptake coefficients for individual SOA species explicitly. The main purpose of this study is to check whether there is any quantifiable effect of $NH_3$ uptake by SOA and

understand how this process could potentially affect air quality predictions. Since the potential impact is significant (as suggested by the results of this study), it should encourage future lab studies to further investigate the mechanism of this process and provide more detailed information about uptake coefficients for more comprehensive modelling studies. More specificity, the statement of "10% of SOA molecules can react with $NH_3$ to form nitrogen‑containing organic compound (NOC)", does not mean only a limiting number of SOA species can react with $NH_3$, but rather indicates the total moles potion of SOA that could potentially interact with $NH_3$. Additionally, the study of Liu et al. (2015) is based on $\alpha$‑pinene and m‑xylene which could not be directly linked with the surrogated SOA species in CMAQ, thus an explicit parameterization of individual SOA species is not possible in this study.

The following statement has been added to the manuscript at Page 4 Line 25, to clarify this point:

*As current laboratory data are not detailed enough to model the chemical uptake of ammonia by individual SOA species explicitly, a range of uptake coefficients were selected and applied to all SOA species. In the future, this approach can be refined by adopting more explicit reactions between ammonia and various types of SOA compounds.*

2. **Model evaluation: this section presents just general PM evaluation and does not have a clear focus on evaluating the parameterization scheme developed. At a minimum, the sites where the model evaluation is based on should be labeled on the model concentration maps from Figure 1 – 6 with the corresponding model biases. The way they are presently listed in Tables is not illustrative and does not help the readers understand the simulation results in the context of model biases. The concentration maps show large spatial variability of the NH3 uptake effect on different aerosol components (e.g. over Southeast US, Central Great Plains), but they are all based on model simulations without observational backup. If there are some observational sites located within those regions where large model sensitivity is found, more discussions of the model bias or improvement after the parameterization scheme should be given to them.**

We thank the reviewer for the excellent suggestion. Additional figures S2‑S11 have been added to the Supporting Information (SI) to illustrate the model bias against observation, as well as how model performance varies for different species at each observation site after the addition of $NH_3$ uptake process. Corresponding discussions have been added in section 3.1 as follows:

Page 5 Line 22: *Additionally, the maps of MNGB values of ozone measured by individual stations are available in the SI section (Figure S2). These maps show that most of the stations have low bias with some underestimation over the north-east in the winter and some overestimation around the country in the summer.*

Page 5 Line 28: *Additionally, maps of MFB values of PM2.5 measured by individual stations are available in the SI section (Figure S3).*

Page 6 Line 2: *with the MFB values for each site mapped in Figure S4.*

Page 6 Line 11: *and the MFB values for ammonia measured by individual stations are presented in Figure S5*

Page 6 Line 13: *especially over the southeast and the Central Valley regions of California.*

Page 7 Line 1: *Figure S6 in the SI section shows the difference of MFE between the base cases and cases with different assumed values for $NH_3$ uptake coefficients. For the winter cases, the overall impact on model performance is negligible. For the summer cases, improvements in model performance can be found in southeast and the Central Valley regions of California. The choice of the $\gamma=10^{-4}$ appears to provide the greatest model performance improvement in the summer, based on both Table 3 and Figure S6.*

Page 7 Line 9: *The statistics for $SO_4^{2-}$ presented in Table S3 of the SI section along with the maps of MFB values for all individual sites (Figure S11) indicate good model performance. There is good agreement between mean observed and simulated concentrations with small MFB and MFE values that satisfy the model performance goals proposed by Boylan and Russell (2006).*

Page 8 Line 7: *Additionally, Figure S7 in the SI section shows the level of bias (MFB) of individual CSN sites for the base case, which shows $NH_4^+$ is considerably overestimated over the southeast but underestimated in the midwest regions of the country for both winter and summer. Based on Table 4, the $NH_4^+$ is slightly overestimated in the base case for the summer period, however, the addition of $NH_3$ uptake leads to a lower modeled $NH_4^+$ concentration and reduced level of overestimation. Such improvements happen over most of the eastern US as well as the Central Valley of California, based on Figure S8 (b) and (d) in the SI which*

*presents the difference in MFE between the base cases and cases with NH₃ uptake coefficients. Similar to NH₃, the choice of the γ=10⁻⁴ appears to provide the greatest model performance improvement in the summer, based on both Table 4 and Figure S8.*

*Page 8 Line 15: As shown on Figures S8 (a) and (c), model performance is not improved in most of the stations, except over the southeast region.*

*Page 8 Line 21: Figure S9 in the SI section shows the maps of MFB values for particulate nitrate measured by each station in the base cases. We find that the modelled $NO_3^-$ is overestimated over the southeast region for both periods, and also overestimated along the Central Valley of California during the summer period. The addition of NH₃ uptake reduced such overestimation and improved the model performance in those regions as shown in Figure S10, which presents the difference of MFE between base cases and cases with different NH₃ uptake coefficients. For the winter period, it is clear the γ=10⁻³ case provides better model performance. For the summer period, the model performance improvement occurred on more observation sites in the γ=10⁻⁴ case than the γ=10⁻³ case. However, the γ=10⁻³ case provides better improvement at some sites, although more sites suffer performance deterioration compares to the γ=10⁻⁴ case.*

3. **Section 3.2.3: the impact of the NH3 uptake on organic PM is large, which apparently contradicts with the earlier claim (pg 4, line 10-15) that the NH3 uptake does not create SOA mass. Later on the authors explained that it was due to changes in C2 aerosol acidity, which in turn drives SOA changes from the acid-catalyzed formation pathways. To avoid confusion, I suggest the authors first present the acidity changes (maybe make it a separate sub-section) and then present the acidity-induced SOA changes. In the SOA section, state upfront that the SOA changes are not caused by the parameterization creating more SOA mass by itself.**

We acknowledge the thoughtful suggestion from the reviewer. The following changes were implemented in the manuscript to clarify this point:

- To resolve the contradiction, a statement has been added at Page 4 Line 18: *Although the NH₃ uptake process does not directly impact the mass of SOA, it can affect the SOA mass indirectly as particle acidity is altered due to this process, which will be discussed in section 3.2.3.*

- The beginning of second paragraph of section 3.2.3 (Page 14 Line 9) was rewritten as follows:

  *As mentioned in section 2, the NH₃ uptake parameterization used in this study does not directly add mass to SOA. However, significant changes in SOA concentration are observed after implementing the NH₃ uptake mechanism, which is indirectly caused by changes in particle acidity discussed later.*

**Technical Comments:**

1. **Equation 1: is the aerosol surface area (Sj) dry or wet area? I think it should be wet area, i.e. considering hygroscopic growth of aerosols under ambient RH conditions. If wet area, how does the model treat SOA hygroscopic growth?**

   This is an excellent point. Indeed, the variable Sj represents the wet area. Only the inorganic hygroscopicity is considered in the current model, with aerosol liquid water (ALW) being calculated using ISORROPIA. Aqueous-phase formation of biogenic SOA from isoprene epoxide processing is the only process that links ALW and SOA in the current model (Pye et al. 2013), where ALW is predicted based on inorganic species can act as a mediator for the SOA formation reaction. In general, the increase of RH and particle-phase ALW content have positive feedback on SOA formation. A direct water uptake to the organic phase has recently been studied by (Pye et al. 2016), but is not yet included in current model.

   We have modified the manuscript to clarify the nature of the surface area at Page 3 Line 30:

   *one must know the representative wet surface area concentration of SOA ($S_{SOA}$) (SOA hygroscopic growth is not considered in the model).*

2. **Pg 4, Line 10: "an SOA compounds" should be "an SOA compound"**

   This error has been corrected as suggested.

[revised manuscript text omitted]

---

## Author Comment (AC2) · 1 Feb 2018

**Interactive Comment reply to Anonymous Referee #2**

Shupeng Zhu[1], Jeremy R. Horne[1], Julia Montoya-Aguilera[2], Mallory L. Hinks[2], Sergey A. Nizkorodov[2], and Donald Dabdub[1]

[1]Computational Environmental Sciences Laboratory, Department of Mechanical & Aerospace Engineering, University of California, Irvine, Irvine, CA, 92697-3975, USA

[2]Department of Chemistry, University of California, Irvine, Irvine, CA, 92697-3975, USA

The authors thank the referee for providing a thorough review and agree that some changes and clarification would improve the manuscript. We would propose to make the revisions outlined below for submission to Atmospheric Chemistry and Physics. Each item starts with the reviewer's comment in bold followed by our response in plain text and blue color. The updated manuscript and supporting information along with the marked-up version of the manuscript are all attached at the end of this response.

**Major Comments:**

1. **The primary limitation of the study is that once NH3 is taken up by SOA in the model, it disappears. As described in the introduction, NH3 can be taken up into SOA by either neutralizing organic acids (producing ammonium salts) or by reacting with aldehyde species to produce NOC (nitrogen-containing organic carbon species, such as imines and imidazoles), most of which are still quite basic and could react with inorganic acids. The relative importance of these two competing reactions is not known, but this study neglects both options. The result is the counterintuitive conclusion that including NH3 uptake to aerosol particles in the model reduces both NH3 (gas) and NH4+ (aerosol) concentrations, while also increasing HNO3 (gas) and decreasing NO3- (aerosol) concentrations. To a great extent, NH3 uptake to SOA must either produce NH4+ in the aerosol particle (by neutralizing an organic acid) or produce basic NOC species that can still neutralize HNO3. NH3 uptake that generates neither of these products, as assumed in this manuscript, does not appear to be a viable option. While the state of knowledge of this chemistry is not quantitative enough to nail this down, and the authors allude to this in the last paragraph in the paper, these issues should be discussed more vigorously in the manuscript.**

   We clarify the assumptions taken in this study. There are three mechanisms by which $NH_3$ can interact with aerosol particles:
   - The most important mechanism, which is implemented in the base version of the model, is formation of inorganic salts of sulfuric and nitric acids. We have

not removed this mechanism from the model and it continues to contribute to PM$_{2.5}$ mass.

- Formation of salts of organic acids is not considered in the base version of the model and is not implemented in the present study.
- The new mechanism added to the model is based on reaction between NH$_3$ and SOA carbonyls. It is correct that we approximate this mechanism by assuming that the gas-phase NH$_3$ taken up by SOA is removed. All the NH$_3$ that is taken up by SOA is considered to be irreversibly transformed into NOC as discussed in section 2 (Page 4 Line 12-16). The resulting NOC is further assumed less able to neutralize inorganic acids compared to NH$_3$, and therefore the ability of NOC to neutralize HNO$_3$ is neglected in this mechanism implemented into the model. We feel this assumption is justified because NH$_3$, with pK$_b$=4.8, is a much more basic than imines (pK$_b$ ~ 10) and nitrogen containing aromatic compounds such as pyrrole (pK$_b$ = 13.6) and pyridine (pK$_b$ = 8.8).

To clarify this point, a new figure (Figure S1) has been added in the Supplement Information (SI) section to illustrate the fate of NH$_3$ in the particle phase. In addition, a statement has been added to Page 4 Line 9 as follows:

*In this study, all NH$_3$ taken up by SOA carbonyls is assumed to form NOCs, such as secondary imines and heteroaromatic compounds (Laskin et al., 2015).*

And added following statement at Page 4 Line 18.

*Although, the NH$_3$ uptake process does not directly impact the mass of SOA, it can affect the SOA mass indirectly as particle acidity is altered due to this process, which will be discussed in section 3.2.3. Figure S1 in the SI section shows a schematic representation of the NH$_3$ reactions considered in the model, including reversible function of inorganic salts and irreversible formation of NOC. The ability of NOCs to neutralize inorganic acids is not considered (see Figure S1.) because NOCs are much weaker bases (e.g., imine pK$_b$ ~ 10, pyrrole pK$_b$ = 13.6, pKb = 8.8) compared to NH$_3$ (pK$_b$ = 4.8). In other words, once NH$_3$ is converted into NOC it is no longer available to make inorganic salts of nitrate and sulfate.*

**Specific Comments:**

1. **It would be helpful to mention whether aerosol in the model are externally or internally mixed.**

   Aerosol in the model is internally mixed. We added the statement in Page 3 line 27:

*The size distribution of the aerosol particles is represented by 3 log-normal modes: the Aitken mode (size up to approximately 0.1 μm), the accumulation mode (size between 0.1 μm to 2.5 μm) and the coarse mode (size between 2.5 μm to 10 μm). The particles are assumed to be internally mixed within each mode.*

2. **p. 11 line 8: This sentence implies that both California's central valley and the South Coast Air Basin have high NH3 emissions from intensive agriculture. Is this really true in the latter case?**

The South Coast Air Basin has high NH₃ emissions from intensive agriculture, especially from dairy facilities located in Chino, CA. Figure 1 shows the number of dairy cows and the aircraft measured NH₃ concentrations (Nowak et al., 2012) in the South Coast Air Basin. This new reference has been added to the manuscript to justify this point in Page 12 Line 16.

[Figure]

Figure 1. Number of dairy cows and the aircraft measured NH₃ concentrations in the South Coast Air Basin.
Ref: Nowak, J. B., J. A. Neuman, R. Bahreini, A. M. Middlebrook, J. S. Holloway, S. A. McKeen, D. D. Parrish, T. B. Ryerson, and M. Trainer. "Ammonia sources in the California South Coast Air Basin and their impact on ammonium nitrate formation." Geophysical Research Letters 39, no. 7 (2012).

3. **p. 11 line 9: Where do organic acids fit in the order of NH3 neutralization with H2SO4 and HNO3? If NH3 uptake to SOA results in neutralization of organic acids, does this affect any of the manuscript's conclusions about HNO3 (g) concentrations increasing and nitrate concentrations decreasing in response to NH3 uptake?**

We acknowledge the reviewer's concern over organic acids. As addressed on the major comments, no direct interaction between organic acids and NH₃ is considered in the model. We do not think organic acids can compete with sulfuric and nitric acid when it comes to neutralizing ammonia.

4. **p. 12 line 4: The prediction of almost no nitrate in summer aerosol over the southeast U.S., due to sulfuric acid neutralizing all of the available NH3, should be testable against regional PM observations. Is the prediction consistent with this dataset?**

   Thank you for pointing this out. This prediction is in fact consistent with the observational data. We added Figure S9 in the SI to show the level of model bias against observation. A small bias is found in the southeast region in the base case, and even stations with large bias indicates an over estimation by the model, which means the observed nitrate concentration is even lower than the model prediction. Several sentences have been added to Page 8 Line 21 as follows:

   *Figure S9 in the SI shows the map of MFB value of each station for the base cases. We find that $NO_3^-$ is overestimated over the southeast region for both periods, and also overestimated along the Central Valley of California during the summer period. The addition of $NH_3$ uptake process eased such overestimation and improved the model performance in those region as shown on Figure S10, which presents the difference of MFE between base cases and cases with $NH_3$ uptake included. For the winter period, it is clear the $\gamma=10^{-3}$ case provides better model performance. For the summer period, the model performance improvement is more wide spread in the $\gamma=10^{-4}$ case than the $\gamma=10^{-3}$ case, while the $\gamma=10^{-3}$ case provides a deeper improvement at some sites with more sties suffer performance deterioration compares to the $\gamma=10^{-4}$ case.*

5. **p. 12 line 9: This sentence is an example of the strange reasoning caused by the lack of a product formed by NH3 uptake in the model. "The reduction in NH3 due to the SOA uptake, directly impacts the available NH3 that could be condensed into the particle phase, and reduces the NH4+ concentration considerably."**

   This comes back to the previously addressed major comment by the reviewer. As addressed in the response to major comments and on Page 4 Line 10-25 of the manuscript, the $NH_3$ uptake by SOA is considered to form NOCs, which cause $NH_3$ and $NH_4^+$ concentrations to decrease simultaneously.

**Technical Corrections:**
1. **p. 3 line 31: the phrase in parentheses does not make sense.**

   We apologize for this confusion. It has been rephrased as follows at Page 3 Line 32:

   *Assuming a uniform density across different chemical components.*

2. **p. 9 line 10: the sentence with the phrase "east remote source and go under. . ." does not make sense. In the following sentence, when the authors write "the introduction of NH3 does not have much impact on this spot" do they mean "the addition of NH3 uptake to the model does not have much impact at this location"?**

   The first point has been rephrased and split into two sentences (Page 11 Line 6):

   *The winter hot spot around northeastern Utah (Uintah Basin) could be caused by the relatively static atmospheric conditions during the winter in the valley (Lee et al., 2014), which traps $NO_x$ emitted from local and remote sources located on the east side of the valley. The resulting $NO_x$ undergoes a nighttime reaction with $O_3$ forming $N_2O_5$ (high $N_2O_5$ concentration were spotted in the same place as shown on Figure S24).*

   For the second point, yes, it means "the addition of $NH_3$", we have rephrased the word accordingly, and replaced "introduction of $NH_3$" by "addition of $NH_3$" throughout the manuscript to avoid vague statements.

3. **p. 11 line 10: the meaning of "association form of NH4+" is unclear.**

   We rephrased it with: "*The percentage of $NH_4^+$ associated with $NO_3^-$, $SO_4^{2-}$ and $HSO_4^-$*" in Page 12 Line 19.

4. **p. 13 line 21: "wide" should be "widespread"**

   It has been corrected accordingly at Page 14 Line 6.

5. **p. 16 line 14: The growth of AISO3 with respect to the uptake coefficient is linear, not exponential, since the uptake coefficients were varied exponentially.**

   We appreciate the reviewer's correction. We have change "exponential" to "linear" in Page 16 Line 4.

6. **p. 20 line 16: the meaning of the parenthetical phrase "based high NOx assumption" is unclear.**

[revised manuscript text omitted]

---

## Author Response (AR2)

**Comment reply to Co-Editor**

Shupeng Zhu[1], Jeremy R. Horne[1], Julia Montoya-Aguilera[2], Mallory L. Hinks[2], Sergey A. Nizkorodov[2], and Donald Dabdub[1]

[1]Computational Environmental Sciences Laboratory, Department of Mechanical & Aerospace Engineering, University of California, Irvine, Irvine, CA, 92697-3975, USA
[2]Department of Chemistry, University of California, Irvine, Irvine, CA, 92697-3975, USA

The authors thank the Co-Editor for providing prompt and thorough advise for the revised paper. We agree that a few more changes and clarification would improve the manuscript. We propose to make the revisions outlined below for submission to Atmospheric Chemistry and Physics. Each item starts with the Co-Editor comment in bold followed by our response in plain text and blue color.

**1) The colorbars in some of the supplementary figures is pretty bad. I am referring to the ones where zero values are black (e.g. S9). Can you use the colorbar used in e.g. S10 instead? Also, for the figures using the S10 colorbar, the saturated colors on both sides are almost the same, can you make them a bit more distinct?**

We acknowledge the thoughtful suggestion from the Co-Editor. We have changed the color-bar in the SI accordingly.

**2) If I understand correctly your answer to the major comment of reviewer 2 (and some of the specific comments come back to this), your statement added in page 4 line 18 "Although, the NH3 uptake process does not directly impact the mass of SOA, it can affect the SOA mass indirectly as particle acidity is altered due to this process, which will be discussed in section 3.2.3. " means that part of NH3 is lost so less is available to form NH4+ which is why acidity is affected. This should be clearly stated here, since a couple sentences later you say that "The ability of NOCs to neutralize inorganic acids is not considered (see Figure S1.)", which is the opposite of what you say in the previous statement, as a standalone sentence. I had to dig deeply in the text of section 3.2.3 to figure this out, can you make it more visible here instead of simply referring to the section?**

We acknowledge the thoughtful suggestion from the Co-Editor. It is correct that the particle acidity is altered as part of $NH_3$ is lost so less is available to form $NH_4^+$. However, it does not actually contradict the statement "The ability of NOCs to neutralize inorganic acids is not considered (see Figure S1.)". This statement is meant to clarify why NOCs themselves do not affect particle acidity, because they are assumed to be much less basic than ammonia in this study. So, the reasoning behind the science is that the loss of $NH_3$ makes NOCs (which cannot neutralize acids) instead of $NH_4^+$. As less $NH_4^+$ is formed and the NOCs that replace $NH_4^+$ are not ionized, the particle acidity increases.

To further clarify this point, we have rewritten the statement on page 4 line 18 as follows:

*Although, the $NH_3$ uptake process does not directly impact the mass of SOA, the conversion of NH3 into NOCs can affect the SOA mass indirectly due to particle acidity changes. The particle acidity is altered because strongly basic $NH_3$, which is converted into weakly-basic NOCs, is no longer available to form inorganic salts of $NH_4^+$. As the extent of neutralization of inorganic acids with $NH_3$ is reduced the particle acidity may increase.*